# FAST HYPERBOLOID DECISION TREE ALGORITHMS

**Philippe Chlenski,**[1] **Ethan Turok,**[1] **Antonio Moretti,**[2] **Itsik Pe'er**[1]
[1]Columbia University [2]Barnard College

## ABSTRACT

Hyperbolic geometry is gaining traction in machine learning due to its capacity to effectively capture hierarchical structures in real-world data. Hyperbolic spaces, where neighborhoods grow exponentially, offer substantial advantages and have consistently delivered state-of-the-art results across diverse applications. However, hyperbolic classifiers often grapple with computational challenges. Methods reliant on Riemannian optimization frequently exhibit sluggishness, stemming from the increased computational demands of operations on Riemannian manifolds. In response to these challenges, we present HYPERDT, a novel extension of decision tree algorithms into hyperbolic space. Crucially, HYPERDTeliminates the need for computationally intensive Riemannian optimization, numerically unstable exponential and logarithmic maps, or pairwise comparisons between points by leveraging inner products to adapt Euclidean decision tree algorithms to hyperbolic space. Our approach is conceptually straightforward and maintains constant-time decision complexity while mitigating the scalability issues inherent in high-dimensional Euclidean spaces. Building upon HYPERDT we introduce HYPERRF, a hyperbolic random forest model. Extensive benchmarking across diverse datasets underscores the superior performance of these models, providing a swift, precise, accurate, and user-friendly toolkit for hyperbolic data analysis. Our code can be found at `https://github.com/pchlenski/hyperdt`.

## 1 INTRODUCTION

### 1.1 BACKGROUND: HYPERBOLIC EMBEDDINGS

The adoption of hyperbolic geometry for graph embeddings has sparked a vibrant and rapidly growing body of machine learning research (Sarkar, 2012; Chamberlain et al., 2017; Gu et al., 2019; Chami et al., 2020; 2021). This surge in interest is driven by the compelling advantages offered by hyperbolic spaces, particularly in capturing hierarchical and tree-like structures inherent in various real-world datasets. In hyperbolic space, neighborhoods grow exponentially rather than polynomially, allowing for embeddings of exponentially growing systems such as phylogenetic trees or concept hierarchies. Hyperbolic embeddings have proven to be highly effective, showcasing state-of-the-art results across various applications including question answering (Tay et al., 2018), node classification (Chami et al., 2020), and word embeddings (Tifrea et al., 2018). These achievements underscore the growing interest in classifiers that operate natively within hyperbolic spaces (Gulcehre et al., 2018; Marconi et al., 2020; Doorenbos et al., 2023).

While hyperbolic classifiers leverage the curvature properties of hyperbolic geometry to make more nuanced and accurate predictions of hierarchical data, such techniques often face a dilemma. Methods employing Riemannian optimization often exhibit sluggishness due to the increased computational complexity associated with operations on Riemannian manifolds, which require intricate geometric calculations. The sensitivity of Riemannian optimization to initialization and the presence of complex geometric constraints can further contribute to slower convergence. Other methods are consistent with hyperbolic geometry but incur time-complexity penalties associated with horosphere calculations (see, for example, Fan et al. (2023); Doorenbos et al. (2023)). Others apply Euclidean methods to hyperbolic data transformed with logarithmic maps (Chami et al., 2019; Chen et al., 2022), whitening techniques (Chami et al., 2021), or directly on hyperbolic coordinates (Jiang et al., 2022). While effective, such methods can ignore the geometric structure of the data or introduce additional complexity to the inference process, adversely affecting both speed and interpretability.

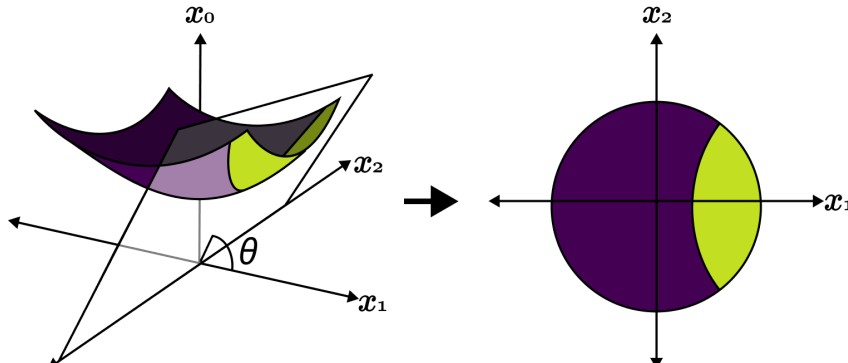

Figure 1: Geodesic partitions in the hyperboloid model $\mathbb{H}^{2,1}$ (left) and poincare model $\mathbb{P}^{2,1}$ (right) into two halves (purple/yellow). In $\mathbb{H}^{2,1}$, a geodesic can be expressed as the intersection of the hyperboloid with an angled plane through the origin of the ambient space (transparent white). While these two representations are equivalent these partitions can be expressed more compactly in $\mathbb{H}^{2,1}$.

## 1.2 The Need for Hyperbolid Decision Trees and Paper Contributions

Decision trees are workhorses of machine learning, favored for a variety of reasons, including:

- *Speed*. Decision trees are relatively fast to train, and efficient prediction makes them suitable for real-time or near-real-time applications.
- *Interpretability*. Decision trees provide a highly interpretable model, unlike random forest methods that can be opaque regarding how features combine to form a predictor.
- *Simplicity*. Decision trees involve straightforward logic, making them easier to implement and explain than multilayered, highly parameterized network models.

The additional complexities introduced by hyperbolic geometry, which often necessitate intricate optimizations over Riemannian manifolds, introduce a disparity between the existing availability of efficient decision tree algorithms tailored to such spaces and their potential transformative impact.

This paper introduces a novel approach to extend traditional Euclidean decision tree algorithms to hyperbolic space. This approach, called HYPERDT, is conceptually straightforward and maintains constant-time decision complexity while mitigating the scalability issues inherent in high-dimensional Euclidean spaces. Building upon HYPERDT, we introduce an ensemble model called HYPERRF, which is an extension of random forests tailored for hyperbolic space. Our contributions in this work are summarized as follows:

1. We develop HYPERDT, a novel extension of decision trees to data in hyperbolic space, e.g. learned hyperbolic embeddings. HYPERDT avoids computationally expensive Riemannian optimization, numerically unstable exponential or logarithmic maps, and quadratically scaling pairwise comparisons between all data points. Instead, we reframe decision trees in Euclidean space in terms of inner products, yielding a natural extension to hyperbolic spaces of arbitrary negative curvature. Like other decision tree algorithms, HYPERDT repeatedly partitions the input space into decision areas, i.e. subspaces labeled with the majority class for the points in the training set for that region. However, because it uses geodesic submanifolds to partition the space, HYPERDT is the first decision tree predictor whose decision areas maintain convexity and topological continuity for arbitrary partitions.

2. We generalize HYPERDT to random forests in a second algorithm which we refer to as HYPERRF. In select cases, HYPERRF demonstrates enhanced accuracy and reduced susceptibility to overfitting when contrasted with HYPERDT.

3. We demonstrate state-of-the-art accuracy and speed of HYPERDT and HYPERRF on classification problems compared to existing counterparts on various datasets.

4. We provide a Python implementation of HYPERDT and HYPERRF for classification and regression following SCIKIT-LEARN API conventions (Pedregosa et al., 2011).

## 1.3 RELATED WORK

Several graph embedding methods have been proposed in the Poincaré disk (Nickel & Kiela, 2017; Chamberlain et al., 2017), hyperboloid model (Nickel & Kiela, 2018), and even in mixed-curvature products of manifolds (Gu et al., 2019). De Sa et al. (2018) provides a thorough overview and comparison of graph embedding methods in terms of their metric distortions.

Hyperbolic embeddings have found diverse applications across domains, particularly in computational biology and concept ontologies, both of which are structured by latent branching relationships. In biology, inheritance patterns are tree-like: at evolutionary scales, hyperbolic embeddings successfully model well-known phylogenetic trees (Hughes et al., 2004; Chami et al., 2020; Jiang et al., 2022). Furthermore, Corso et al. (2021) learn faithful hyperbolic species embeddings directly from nucleotide sequences, bypassing phylogenetic tree construction. On shorter timescales, such as single-cell RNA sequencing data, where cell types evolve from progenitors, a tree-like structure emerges once more. Ding & Regev (2021) showcased that variational autoencoders with a hyperbolic latent space effectively capture the branching patterns of cells' developmental trajectories.

For concept ontologies like WordNet, hyperbolic embeddings effectively capture subclass relationships among nouns, improving link prediction accuracy (Ganea et al., 2018). Additionally, Tifrea et al. (2018) showcase the utility of hyperbolic embeddings for unsupervised learning of latent hierarchies when explicit ontologies are absent, a capability possibly underpinned by the hierarchical nature of concepts within WordNet. Moreover, a recent study by Desai et al. (2023) unveils that text-image representations in hyperbolic space exhibit enhanced interpretability and structural organization while maintaining performance excellence in standard multimodal benchmarks.

Machine learning on hyperbolic embeddings is an emerging and evolving research area. Cho et al. (2018) and (Fan et al., 2023) have proposed approaches for adapting support vector machines to hyperbolic space, which has proven useful in biological contexts (Agibetov et al., 2019). In the realm of neural methods, there have been developments such as hyperbolic attention networks (Gulcehre et al., 2018), fully hyperbolic neural networks Chen et al. (2022), and hyperbolic graph convolutional networks Chami et al. (2021). A recent contribution by Doorenbos et al. (2023) introduced HORORF, a variant of the random forest method that employs decision trees with horospherical node criteria. Our approach is most similar to Doorenbos et al. (2023); however, it differs in that it utilizes geodesics as opposed to horospheres and does not require pairwise comparisons between data points, leading to an algorithm that maintains constant time decision complexity.

Improving decision tree and random forest algorithms has also attracted considerable research attention, although except for HORORF, none of these methods are designed for use with hyperbolic embeddings. In particular, improvements like gradient boosting Chen & Guestrin (2016) and optimization based on branch-and-bound methods Lin et al. (2022); McTavish et al. (2022); Mazumder et al. (2022) have been proposed to circumvent some of the suboptimal qualities of classic CART decision trees Breiman (2017), which uses a greedy heuristic to select each split in the tree.

## 2 PRELIMINARY

### 2.1 HYPERBOLIC SPACES

Hyperbolic geometry, characterized by its constant negative curvature, can be represented by various models, including the hyperboloid (also known as the Lorentz or Minkowski model), the Poincaré disk model, the Poincaré half-plane model, and the (Beltrami-)Klein model. We use the hyperboloid model due to its simplicity in expressing geodesics using plane geometry, which we exploit to define our decision boundaries. While prior research in this field has predominantly centered on the Poincaré disk model, the straightforward conversion between Poincaré disk coordinates and hyperboloid coordinates (See Section A.1 for details) allows for seamless integration of techniques across different hyperbolic representations, a flexibility we leverage in our work.

The $D$-dimensional hyperboloid model is embedded inside an ambient $(D + 1)$-dimensional Minkowski space, a metric space equipped with the Minkowski inner product:

$$\langle \mathbf{x}, \mathbf{x}' \rangle_{\mathcal{L}} = -x_0 x_0' + \sum_{i=1}^{D} x_i x_i'. \tag{1}$$

The above is equivalent to an adaptation of the Euclidean inner product, where the first term is negated. This distinguished first dimension, often termed the "timelike" dimension, earns its name due to its significance in the context of special relativity theory.

Let the hyperboloid have constant negative scalar curvature $-K$. Inside the Minkowski space, points are assumed to lie on $\mathbb{H}^{D,K}$, the hyperboloid of dimension $D$ and curvature $K$:

$$\mathbb{H}^{D,K} = \left\{ \mathbf{x} \in \mathbb{R}^{D+1} : \langle \mathbf{x}, \mathbf{x} \rangle_{\mathcal{L}} = -1/K, \ x_0 > 0 \right\} . \tag{2}$$

That is, the hyperboloid model assumes points lie on the surface of the upper sheet of a two-sheeted hyperboloid embedded in Minkowski space (see Figure 1). The distance between two points on $\mathbb{H}^{D,K}$ is

$$\delta(\mathbf{x}, \mathbf{x}') = \cosh^{-1}(-K \langle \mathbf{x}, \mathbf{x}' \rangle_{\mathcal{L}})/\sqrt{K}. \tag{3}$$

This distance can be interpreted as the length of the geodesic, the shortest path on the manifold connecting $x$ and $x'$. In the hyperboloid model, all geodesics are intersections of $\mathbb{H}^{D,K}$ with respective 2D planes that pass through the origin of the Minkowski space.

## 2.2 DECISION TREE ALGORITHMS

The *Classification and Regression Trees* (CART) algorithm is a mainstay of machine learning, alongside extensions such as random forests (Breiman, 2001) and XGBoost (Chen & Guestrin, 2016). CART recursively partitions the feature space into increasingly homogenous subspaces by maximizing the information gain at each split $S$, $IG(S)$. It measures improved homogeneity due to splitting a dataset $\mathbf{X}$ into subsets $\mathbf{X}^0, \mathbf{X}^1$ of respective fractions $f^i = |\mathbf{X}^i|/\mathbf{X}$:

$$IG(S) = C(\mathbf{X}) - f^0 C(\mathbf{X}^0) - f^1 C(\mathbf{X}^1), \tag{4}$$

where $C(\cdot)$ is the impurity or cost function of each set. Objective functions like Gini impurity, mean squared error (MSE), and entropy are popular choices for $C(\cdot)$. The tree is iteratively constructed until further splits will constitute overfitting. A decision tree is often used as is to make predictions. The *Random Forest* algorithm integrates predictions across ensembles of decision trees trained on randomly subsampled subsets of the training data using a majority voting procedure.

## 3 HYPERBOLOID DECISION TREE ALGORITHMS

Extending CART to hyperbolic space involves several essential steps. First, as outlined in Section 3.1, we express Euclidean decision boundaries in terms of inner products, providing the requisite geometric intuition for hyperbolic decision trees. In Section 3.2, we utilize these inner products to establish a streamlined decision process based on geodesic submanifolds in hyperbolic space. Section 3.3 discusses the selection of candidate hyperplanes, Section 3.4 presents closed-form equations for decision boundaries, and Section 3.5 describes the Hyperbolic Random Forest extension HYPERRF.

## 3.1 FORMULATING DECISION TREES WITH INNER PRODUCTS

Traditionally, a split is perceived as a means to ascertain whether the value of a point $\mathbf{x}$ in a given dimension $d$, is greater or lesser than a designated threshold, $\theta$, namely,

$$S(\mathbf{x}) = \mathbb{I}\{x_d > \theta\} . \tag{5}$$

This decision boundary can also be thought of as the axis-parallel hyperplane $x_d = \theta$, and thus we can rewrite the same split as follows

$$S(x) = \max(0, \ \text{sign}(\mathbf{x} \cdot \mathbf{n}(d) - \theta)) , \tag{6}$$

where $\mathbf{n}(d)$ is the one-hot base vector along dimension $d$, i.e. the normal vector of our decision hyperplane. Of course, Eq. 5 is the practical, $O(1)$ condition. The slower, $O(D)$ Eq. 6 is instructive towards the hyperboloid generalization, and can still be computed in $O(1)$ due to sparsity of $\mathbf{n}(d)$.

## 3.2 Extension to the Hyperboloid Model

We will now modify decision tree splits for hyperbolic space. Splitting a decision tree along standard axis-aligned hyperplanes is inappropriate when all the datapoints lie on the hyperboloid: the intersection of an axis-aligned $D+1$ dimensional hyperplane with the hyperboloid $\mathbb{H}^{D,K}$ in $(D+1)$-dimensional Minkowski space results in a $D$-dimensional hyperbola, which lacks any meaningful interpretation within the hyperboloid model. Euclidean CART generates such decision boundaries, but they are likely ill-suited to capture the geometry of hyperbolic space.

On the other hand, $D$-dimensional homogeneous hyperplanes, i.e. hyperplanes that contain the origin, intersect $\mathbb{H}^{D,K}$ as geodesic submanifolds. In 3D Minkowski space, geodesics between any $\{\mathbf{x}, \mathbf{x}'\} \subset \mathbb{H}^{2,1}$ lie on the intersection of $\mathbb{H}^{2,1}$ with some homogeneous 2D plane, as in Figure 1. In higher dimensions, homogeneous hyperplanes intersect $\mathbb{H}^{D,K}$ as $(D-1)$-dimensional geodesic submanifolds, which likewise contain all geodesics between their elements. Partitions by homogeneous hyperplanes maintain convexity and topological continuity: all pairs of points in a subspace are reachable by shortest paths that stay completely within their own subspace.

Building upon the inner product formulation of splits Euclidean CART (Eq. 6), we can substitute a set of geometrically appropriate decision boundaries without altering the rest of the CART framework. Specifically, we replace axis-parallel hyperplanes with homogeneous ones.

To maintain the dimension-by-dimension character of Euclidean CART and enforce sparse normal vectors for efficient inner product calculations, we further restrict the number of decision boundary candidates to $O(D|\mathbf{X}|)$ by only considering rotations of the plane $x_0 = 0$ along a single other axis $d$. These hyperplanes are fully parameterized by $d$ and the rotation angle $\theta$, yielding corresponding normal vectors

$$\mathbf{n}(d, \theta) := \langle n_0 = -\cos(\theta), 0, \ldots, 0, n_d = \sin(\theta), 0, \ldots, 0 \rangle. \tag{7}$$

They define hyperplanes that satisfy

$$x_0 \cos(\theta) - x_d \sin(\theta) = 0 \tag{8}$$

The sparsity of $\mathbf{n}(d, \theta)$ yields a compact $O(1)$ decision procedure:

$$S(x) = \text{sign}\left(\max\left(0, \ \sin(\theta)x_d - \cos(\theta)x_0\right)\right) \tag{9}$$

Notably, this procedure determines points' position relative to a geodesic decision boundary without computing the actual location of the geodesic on $\mathbb{H}^{D,K}$. Because of this, it is also curvature-agnostic. Hyperbolic decision trees compose splits analogously to Euclidean CART: the same objective functions are applicable, and so is the consideration of a single candidate decision boundary per point per (space-like) dimension, resulting in identical asymptotic complexity.

## 3.3 Choosing Candidate Hyperplanes

In Euclidean CART, candidate thresholds are classically chosen among midpoints between successive observed $x_d$ values in the data. The hyperbolic case is slightly more nuanced.

Each decision boundary for dimension $d$ is parameterized by an angle $\theta$, instead of a coordinate value. Each point $\mathbf{x}$ lies on a plane of angle $\theta = \tan^{-1}(x_0/x_d)$. The midpoint angle $\theta_m$ between two angles $\theta_1 < \theta_2$ is defined in terms of points lying on the intersection of these hyperplanes with $\mathbb{H}^{D,K}$. By setting all dimensions besides 0 and $d$ to zero, we can solve for the angle corresponding to the point on $\mathbb{H}^{D,K}$ that is exactly equidistant to the points corresponding to angles $\theta_1, \theta_2$:

$$\theta_m = \begin{cases} \cot^{-1}\left(V - \sqrt{V^2 - 1}\right) & \text{if } \theta_1 < \pi - \theta_2 \\ \cot^{-1}\left(V + \sqrt{V^2 - 1}\right) & \text{if } \theta_1 > \pi - \theta_2 \end{cases} \tag{10}$$

where $V := \frac{\sin(2\theta_1 - 2\theta_2)}{2\sin(\theta_1 + \theta_2)\sin(\theta_2 - \theta_1)}$. See Appendix Section A.3 for a full derivation.

## 3.4 Parameterizing Decision Boundaries

Let $\mathbf{P}(\theta)$ be a decision hyperplane learned by HYPERDT (without loss of generality, assume $d = 1$). We derive closed-form equations for the geodesic submanifold where $\mathbf{P}(\theta)$ intersects $\mathbb{H}^{D,K}$. When all other dimensions are 0, this occurs when

$$x_0 = \alpha(\theta, K) \sin(\theta); \ x_1 = \alpha(\theta, K) \cos(\theta), \tag{11}$$

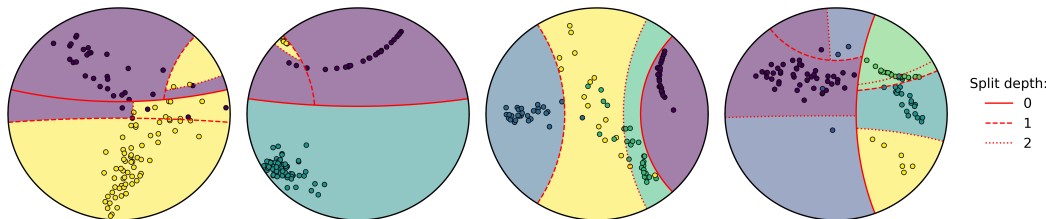

Figure 2: Learned HYPERDT decision boundaries for 2, 3, 4, and 5-class mixtures of wrapped normal distributions visualized on the Poincaré disk. All trees have a maximum depth of 3 and forgo post-training pruning. In the visualization, regions are colored according to their *predicted* class labels while data points are colored according to their *true* class labels.

where $\alpha(\theta, K) := \sqrt{-\sec(2\theta)}/\sqrt{K}$. Note that this is also the point on the intersection of $\mathbf{P}(\theta)$ and $\mathbb{H}^{D,K}$ that is closest to the origin. We use this to parameterize the entire geodesic submanifold $\mathbf{G^d}$ resulting from intersecting the plane with $\mathbb{H}^{D,K}$:

$$\mathbf{v^0} = \langle \sin(\theta),\ \cos(\theta),\ 0,\ \ldots \rangle \tag{12}$$

$$\mathbf{u^d} = \langle 0,\ \ldots,\ u_d^d = 1,\ \ldots,\ 0 \rangle,\ 2 \leq d \leq D \tag{13}$$

$$\mathbf{G^1}(\theta, K) = \left\{ \cosh(t) \cdot \alpha(\theta, K) \cdot \mathbf{v^0} + \sinh(t)\mathbf{u^2}/\sqrt{K} : t \in \mathbb{R} \right\} \tag{14}$$

$$\mathbf{G^d}(\theta, K) = \left\{ \cosh(t)\mathbf{v^{d-1}} + \sinh(t)\mathbf{u^{d+1}}/\sqrt{K} : \mathbf{v^{d-1}} \in \mathbf{G^{d-1}}(\theta, K),\ t \in \mathbb{R} \right\} \tag{15}$$

For visualization, we use $\mathbb{H}^{2,1}$ projected to the Poincaré disk $\mathbb{P}^{2,1}$. We recursively partition the space, plotting decision boundaries at each level and coloring the partitioned space by the majority class. Figure 2 shows an example of such a plot. See Appendix Section A.2 for a full derivation.

## 3.5 HYPERBOLOID RANDOM FORESTS

Analogously to the Euclidean case, creating a hyperboloid random forest is possible by training an ensemble of hyperboloid decision trees on randomly resampled versions of the training data. For speed, we implement HYPERRF, a multithreaded version of hyperboloid random forests wherein each tree is trained as a separate process.

## 4 CLASSIFICATION EXPERIMENTS

### 4.1 PERFORMANCE BENCHMARK BASELINES

For decision trees, we compare our method to standard Euclidean decision trees as implemented in SCIKIT-LEARN (Pedregosa et al., 2011). Since we use the same parameters as the SCIKIT-LEARN decision tree and random forest classes, we can ensure that implementation details like maximum depth and number of points in a leaf node can be standardized. We additionally compare our random forest method to HORORF (Doorenbos et al., 2023), another ensemble classifier for hyperbolic data.

Since HORORF does not implement a single decision tree method, we modified its code to avoid resampling the training data when training a single tree. We call this version HORODT.

For all predictors, we use trees with depth $\leq 3$ and $\geq 1$ sample per leaf. For random forests, all methods use an ensemble of 12 trees. We explore performance in $D = 2, 4, 8,$ and 16 dimensions.

### 4.2 DATASETS

**Synthetic Datasets.** We create a hyperbolic mixture of Gaussians, the canonical synthetic dataset for classification benchmarks, following Cho et al. (2018). We use the wrapped normal distribution on the hyperboloid for each Gaussian as described in Nagano et al. (2019). We draw the means of each Gaussian component from a normal distribution in the tangent plane at the origin and project it onto the hyperboloid directly using an exponential map. The timelike components of random

Gaussians may grow quite large in high dimensions, creating both numerical instability and trivially separable clusters. We thus shrink the covariance matrix $D$-fold. See Section A.4 for details.

**NeuroSEED RNA Embeddings.** NeuroSEED (Corso et al., 2021) is a method for embedding DNA sequences into a (potentially hyperbolic) latent space by using a Siamese neural network with a distance-preserving loss function. This method encodes DNA sequences directly into latent space without first constructing a phylogenetic tree—or, equivalently, it embeds a dense graph of pairwise edit distances. We trained 2, 4, 8, and 16-dimensional Poincaré embeddings of the 1,262,987 16S ribosomal RNA sequences from the Greengenes database (McDonald et al., 2023), then filtered them to the 37,215 that have been identified in the American Gut Project McDonald et al. (2018). This downsampling yields a clinically relevant subset of all Prokaryote species. For the purposes of these benchmarks, however, we restricted ourselves to predicting the six most abundant phyla: Firmicutes, Proteobacteria, Bacteroidetes, Actinobacteria, Acidobacteria, and Plantomycetes.

**Polblogs Graph Embeddings.** We use Polblogs (Adamic & Glance, 2005), a canonical dataset in the hyperbolic embeddings literature for graph embeddings. In the Polblogs dataset, nodes represent political blogs during the 2004 United States presidential election, and edges represent hyperlinks between blogs. Each blog is labeled according to its political affiliation, "liberal" or "conservative." We use the `hypll` (van Spengler et al., 2023) Python implementation of the Nickel & Kiela (2017) method to compute 10 randomly initialized Poincaré disk embeddings in 2, 4, 8, and 16 dimensions.

## 4.3 BENCHMARKING PROCEDURE

We benchmarked our method against Euclidean random forests as implemented in SCIKIT-LEARN, and against HORORF. Each predictor was run with the same settings, including dataset and random seed. HORORF and HORODT used Poincaré disk coordinates and all other models used hyperboloid model coordinates. Each dataset was converted to the appropriate model of hyperbolic space for its predictor before training.

For Gaussian and NeuroSEED datasets, we drew 100, 200, 400, and 800 samples using the same five seeds. We recorded micro- and macro-averaged F1 scores, AUPRs, and run times under 5-fold cross-validation. Cross-validation instances were seeded identically across predictors. We could not produce per-fold timing for HORORF, so instead we recorded the time it took to run the full 5-fold cross-validation. For fairness, we timed the data loading scripts separately and subtracted these times from the reported HORORF times.

We conducted paired, two-tailed t-tests comparing each pair of predictors and marked significant differences in Table 1. A full table of $p$-values is available in Section A.7 in the Appendix.

Benchmarks were conducted on an Ubuntu 22.04 machine equipped with an Intel Core i7-8700 CPU (6 cores, 3.20 GHz), an NVIDIA GeForce GTX 1080 GPU with 11 GiB of VRAM, and 15 GiB of RAM. Storage was handled by a 2TB HDD and a 219GB SSD. Experiments were implemented using Python 3.11.4, accelerated by CUDA 11.4 with driver version 470.199.02.

## 4.4 RESULTS

**Classification Scores.** The results of the classification benchmark are summarized in Table 1. Out of 36 distinct dataset, dimension, and sample size combinations, HYPERDT had the highest score 28 times (one of which was a tie with SCIKIT-LEARN). We demonstrated a statistically significant advantage over SCIKIT-LEARN decision trees in 7 cases, and over HoroRF in 27 cases. SCIKIT-LEARN statistically outperformed HYPERDT once, on the 8-dimensional Polblogs dataset.

Similarly, HYPERRF won 22 times, tying once each with SCIKIT-LEARN and HoroRF, and statistically outperforming SCIKIT-LEARN in 11 cases and HORORF in 13 cases. SCIKIT-LEARN statistically outperformed HYPERRF in 4 cases, all on 8-dimensional NeuroSEED data, and HoroRF statistically outperformed HYPERRF in 2 cases, both on 16-dimensional NeuroSEED data.

Overall, both HYPERDT and HYPERRF showed substantial advantages over comparable methods on the datasets and hyperparameters tested. In high dimensions, classifiers tended to converge to uniformly high performance. The best model for NeuroSEED embeddings varied by dimensionality, a phenomenon that warrants further investigation.

| Data | $D$ | $n$ | Decision Trees | | | Random Forests | | |
|---|---|---|---|---|---|---|---|---|
| | | | HyperDT | Sklearn | HoroDT | HyperRF | Sklearn | HoroRF |
| Gaussian | 2 | 100 | **89.10**$^\dagger$ | 87.90 | 84.60 | **90.70**$^{\ddagger\dagger}$ | 87.50 | 86.30 |
| | | 200 | **90.05**$^\dagger$ | 89.55 | 84.60 | **90.60** | 89.15 | 89.10 |
| | | 400 | **90.97**$^{\ddagger\dagger}$ | 89.53 | 85.55 | **91.32**$^{\ddagger\dagger}$ | 89.00 | 88.88 |
| | | 800 | **91.88**$^{\ddagger\dagger}$ | 90.14 | 85.75 | **91.99**$^{\ddagger\dagger}$ | 89.33 | 89.45 |
| | 4 | 100 | **98.70**$^\dagger$ | 97.70 | 93.60 | **98.40** | 97.90 | 97.90 |
| | | 200 | **98.75**$^{\ddagger\dagger}$ | 98.10 | 95.80 | **98.85**$^{\ddagger\dagger}$ | 97.90 | 98.05 |
| | | 400 | **99.25**$^{\ddagger\dagger}$ | 98.25 | 96.92 | **99.30**$^{\ddagger\dagger}$ | 98.22 | 98.50 |
| | | 800 | **99.30**$^{\ddagger\dagger}$ | 98.36 | 97.27 | **99.36**$^{\ddagger\dagger}$ | 98.21 | 98.76 |
| | 8 | 100 | **99.70**$^\dagger$ | 99.60 | 97.70 | **99.70** | 99.50 | 99.10 |
| | | 200 | **99.65**$^\dagger$ | 99.60 | 98.20 | **99.75** | 99.70 | **99.75** |
| | | 400 | **99.90**$^\dagger$ | 99.88 | 99.10 | 99.88 | **99.93** | 99.88 |
| | | 800 | **99.96**$^\dagger$ | 99.90 | 99.38 | **99.96** | 99.91 | 99.94 |
| | 16 | 100 | **99.80**$^\dagger$ | 99.50 | 98.80 | **99.80** | 99.60 | 99.60 |
| | | 200 | 99.95 | **100.00**$^\dagger$ | 99.50 | 99.90 | **99.95** | 99.80 |
| | | 400 | **100.00**$^\dagger$ | 99.97 | 99.90 | **100.00** | **100.00** | 99.95 |
| | | 800 | **100.00** | 99.99 | 99.90 | **100.00** | 99.99 | 99.92 |
| NeuroSEED | 2 | 100 | **56.60**$^\dagger$ | 55.60 | 49.70 | **57.20** | 55.70 | 56.80 |
| | | 200 | **59.60**$^{\ddagger\dagger}$ | 58.45 | 50.35 | 60.10 | 58.20 | **60.25** |
| | | 400 | **61.78**$^{\ddagger\dagger}$ | 61.00 | 50.62 | **61.58**$^{\ddagger\dagger}$ | 59.47 | 59.33 |
| | | 800 | **61.69**$^\dagger$ | 61.68 | 54.11 | **62.05**$^{\ddagger\dagger}$ | 59.75 | 59.94 |
| | 4 | 100 | **80.40**$^\dagger$ | 80.30 | 53.40 | **80.90**$^\dagger$ | 79.20 | 71.50 |
| | | 200 | 83.60 | **83.70**$^\dagger$ | 52.70 | **84.45**$^{\ddagger\dagger}$ | 82.00 | 70.40 |
| | | 400 | **83.88**$^\dagger$ | 83.83 | 54.08 | **84.65**$^{\ddagger\dagger}$ | 82.33 | 65.20 |
| | | 800 | 84.49 | **84.50**$^\dagger$ | 55.69 | **84.96**$^{\ddagger\dagger}$ | 82.03 | 65.70 |
| | 8 | 100 | 73.80 | **74.00**$^\dagger$ | 52.60 | 79.50 | **82.80**$^{*\dagger}$ | 70.80 |
| | | 200 | 78.30 | **78.40**$^\dagger$ | 55.35 | 81.40 | **84.35**$^{*\dagger}$ | 65.55 |
| | | 400 | 79.45 | **79.57**$^\dagger$ | 52.30 | 82.42 | **86.40**$^{*\dagger}$ | 62.88 |
| | | 800 | **80.76**$^\dagger$ | **80.76**$^\dagger$ | 50.55 | 82.03 | **86.34**$^{*\dagger}$ | 57.69 |
| | 16 | 100 | **74.10**$^\dagger$ | 73.50 | 61.80 | 80.70 | 79.90 | **83.30** |
| | | 200 | **75.90**$^\dagger$ | 75.55 | 66.75 | 82.70 | 82.55 | **83.85** |
| | | 400 | **77.05**$^\dagger$ | 77.03 | 68.80 | 82.30 | 84.73 | **85.90**$^*$ |
| | | 800 | 79.21 | **79.22**$^\dagger$ | 69.59 | 82.44 | 84.44 | **85.49**$^*$ |
| Polblogs | 2 | 979 | **71.04**$^\dagger$ | 70.35 | 64.73 | 71.40 | **71.65**$^\dagger$ | 66.33 |
| | 4 | 979 | **71.53**$^\dagger$ | 70.83 | 62.38 | **72.31**$^\dagger$ | 72.10 | 68.53 |
| | 8 | 979 | 74.02 | **74.85**$^{*\dagger}$ | 61.58 | 74.87 | **75.36**$^\dagger$ | 63.60 |
| | 16 | 979 | **75.06**$^\dagger$ | **75.05**$^\dagger$ | 63.70 | 76.36 | **76.80**$^\dagger$ | 69.12 |

Table 1: Mean micro-F1 scores for classification benchmarks over 10 seeds and 5 folds. The highest-scoring decision tree and random forests are bolded separately. $*$ means a predictor beat HyperRF, $\dagger$ means a predictor beat HoroRF, and $\ddagger$ means a predictor beat Scikit-learn, with $p < 0.05$.

**Runtime Analysis.** In addition to accuracies, we report runtimes for each classifier on each task. In particular, we are interested in the asymptotic behavior of our predictors as a function of the number of samples being considered. These runtimes are plotted by dataset in Figure 3. We demonstrate that our method, while slower than the Scikit-learn implementation by a constant factor, is always faster than HoroRF and grows linearly in runtime with the number of samples. In contrast, HoroRF grows quadratically in runtime as the number of samples increases. This is likely due to the additional complexity of learning horospherical decision boundaries using HoroSVM.

Because HoroRF is optimized for GPU, whereas HyperRF is optimized for parallelism on CPU, exact runtime ratios can be misleading and highly machine-dependent. Similarly, HyperRF may

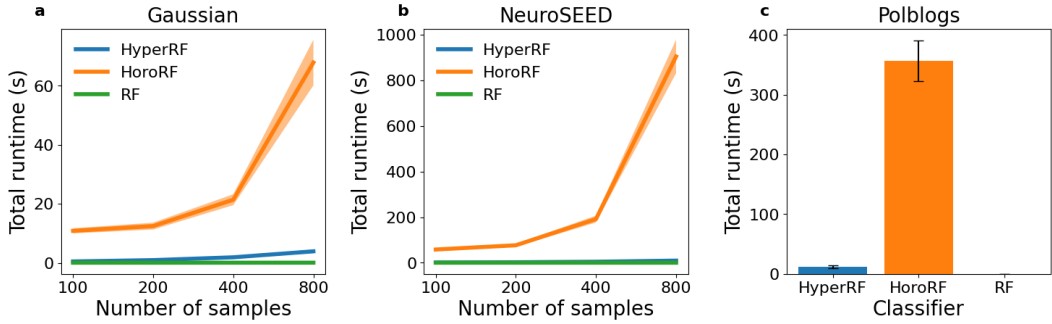

Figure 3: Time to run 5-fold cross-validation, averaged over 10 seeds for each classifier as a function of the number of points. Shaded regions are 95% confidence intervals. Split by dataset: **(a)** wrapped normal mixture, **(b)** NeuroSEED OTU embeddings, and **(c)** Polblogs embeddings.

lack some optimizations found in SCIKIT-LEARN. Therefore, we emphasize the asymptotic aspect of this benchmark, which is agnostic to hardware details and constant-time optimizations.

**Additional experiments**   The results of additional experiments can be found in the Appendix. Section A.6.1 contains additional scaling benchmarks. Sections A.6.2 and A.6.3 extend benchmarks to other hyperbolic classifiers and other models of hyperbolic geometry, respectively. Sections A.6.4 and A.6.5 test performance on image and text embeddings, respectively. Finally, Section A.6.6 tests the impact of ablating the midpoint computation described in Equation 10.

## 5   CONCLUSION

We have introduced HYPERDT, a novel formulation of decision tree algorithms tailored for hyperbolic spaces. This approach leverages inner products to establish a streamlined decision procedure via geodesic submanifolds. HYPERDT exhibits constant-time evaluation at each split and does not rely on Riemannian optimization nor pairwise point comparisons in training or prediction. We extended this technique to random forests with HYPERRF, providing versatile tools for classification and regression tasks. HYPERDT is more accurate than analogous methods in both Euclidean and hyperbolic spaces, while maintaining asymptotic complexity on par with Euclidean decision trees.

The methodological innovation centers on selecting the appropriate decision boundary element to substitute for axis-parallel hyperplanes in the hyperbolic space context. Remarkably, homogeneous hyperplanes serve as highly effective building blocks, preserving continuity and convexity of subspaces at each partition and deviating from traditional approaches by offering straightforward expressions that avoid Riemannian optimization. The trick of using single-axis rotations of the base plane further simplifies and speeds up computation.

HYPERDT and HYPERRF stand out for their speed, simplicity, and stability. The hyperboloid model uses a small, constant number of simple trigonometric expressions at each decision tree node, thereby minimizing numerical instability concerns arising from floating-point issues. Finally, the alignment of hyperboloid geometry to the tree-like structural features of hierarchy-oriented data manifests in single trees performing extraordinarily well, reducing the need for ensemble methods. Such single trees are faster to learn and use. More importantly, they offer full interpretability.

We offer an implementation adhering to standard SCIKIT-LEARN API conventions, ensuring ease of use. Future research avenues can expand HYPERRF with popular features such as gradient boosting, optimization enhancements (e.g., pruning), and additional applications for classification and regression within hyperbolic space. Additional optimizations can greatly improve both performance and usability e.g. through optimizing performance per the standards of SCIKIT-LEARN. Furthermore, new advanced decision tree methods such as Lin et al. (2022), McTavish et al. (2022), and Mazumder et al. (2022), which are based on axis-parallel hyperplanes but apply non-greedy optimizations over multiple splits, can be reformulated in terms of dot-products and applied to homogeneous hyperplanes instead.

ACKNOWLEDGMENTS

We acknowledge the support of the NSF Graduate Research Fellowship under grant no. DGE-2036197 to Philippe Chlenski. We also thank Quentin Chu and Swati Negi for their early use and feedback on our software, which has been instrumental in its development.

ETHICS STATEMENT

This paper aims to advance the field of Machine Learning, conscious of its potential societal impacts and committed to adhering to the ICLR Code of Ethics. Although our research does not directly tackle sensitive ethical issues and we identify no specific societal consequences requiring individual emphasis, we recognize our work within the evolving landscape of machine learning ethics. Acknowledging that the ethical dimensions of machine learning are an area of ongoing exploration and debate, we commit to engaging responsibly with these broader considerations and adhering to ICLR guidelines to address any emerging concerns.

REPRODUCIBILITY STATEMENT

To ensure the reproducibility of our work, we have taken comprehensive steps detailed across our paper, its appendices, and the supplemental GitHub repository. The main text delineates the methodologies and experimental setups, with proofs and derivations of nontrivial mathematical insights in the Appendix. Our GitHub repository houses all of the code, data processing steps, and additional documentation that support the empirical results presented. The `README.md` file for our GitHub repo contains up-to-date, detailed information on which files and notebooks reproduce which parts of the paper and links to data that is not publically available.

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

# A  APPENDIX

## A.1  CONVERSION BETWEEN HYPERBOLOID AND POINCARÉ MODELS

Letting $\mathbf{x_P}$ be a point in $\mathbb{P}^{D,K}$ and $\mathbf{x_H}$ be its equivalent in $\mathbb{H}^{D,K}$,

$$x_{P,i} = \frac{x_{H,i}}{\sqrt{K} + x_{H,0}} \tag{16}$$

$$x_{H,0} = \sqrt{K} \cdot \frac{1 + \|x_P\|_2^2}{1 - \|x_P\|_2^2} \tag{17}$$

$$x_{H,i} = \sqrt{K} \cdot \frac{2x_{P,i}}{1 - \|x_P\|_2^2}. \tag{18}$$

## A.2  GEODESIC SUBMANIFOLD DETAILS

In this section, we will give a full parameterization of the geodesic submanifolds created by intersecting the decision hyperplanes learned by HYPERDT and HYPERRF with $\mathbb{H}^{D,K}$. We enumerate the basis vectors of our decision hyperplanes using a formalism conducive to the following steps, calculate a scaling factor that ensures a basis vector reaches the surface of the manifold, construct a 1-dimensional geodesic arc from the basis vectors of a decision hyperplane, and finally recursively extend this arc to higher dimensions, culminating in a full $(D-1)$-dimensional geodesic submanifold.

To reduce the complexity of notation necessitated by indexing over dimensions, we will assume without loss of generality that the decision hyperplane's normal vector is nonzero in dimensions 0 and 1. We extend this convention to geodesic submanifolds by putting nonzero dimensions first: therefore, we first parameterize an arc along dimension 2, then turn it into a 2-dimensional submanifold along dimension 3, and so on. In general, a $(d \leq D)$-dimensional submanifold will be nonzero in the first $d + 1$ ambient dimensions.

### A.2.1  BASIS VECTORS OF DECISION HYPERPLANES

Let $\mathbf{P}(\theta)$ be a $D$-dimensional decision hyperplane learned by HYPERDT. By our assumption above and Equation 7, the normal vector $\mathbf{n}(1, \theta)$ of $\mathbf{P}(\theta)$ is nonzero only in dimensions 0 and 1. Therefore $\mathbf{P}(\theta)$ has $D$ basis vectors:

$$\mathbf{v^0} = \langle \sin(\theta),\ \cos(\theta),\ 0,\ \ldots \rangle \tag{19}$$

$$\mathbf{u^d} = \langle 0,\ \ldots,\ u_d^d = 1,\ \ldots,\ 0 \rangle,\ 2 \leq d \leq D \tag{20}$$

The $\mathbf{u^d}$ vectors are standard basis vectors which are 1 in dimension $d$ and 0 elsewhere.

### A.2.2  COMPUTING VECTOR SCALE

For what scaling factor $\alpha$ does $\alpha \mathbf{v^0}$ lie on the manifold? We know that $\alpha \mathbf{v^0}$ will always be zero in dimensions 2 through $D$, effectively reducing this to a simple 2-dimensional problem on $\mathbb{H}^{2,K}$:

$$x_1^2 - x_0^2 = -1/K \tag{21}$$

$$x_1^2 = -1/K + x_0^2 \tag{22}$$

$$\alpha^2 \cos^2(\theta) = -1/K + \alpha^2 \sin^2(\theta) \tag{23}$$

$$\alpha^2(\cos^2(\theta) - \sin^2(\theta)) = -1/K \tag{24}$$

$$K\alpha^2 = \frac{-1}{\cos^2(\theta) - \sin^2(\theta)} \tag{25}$$

$$K\alpha^2 = \frac{-1}{\cos(2\theta)} \tag{26}$$

$$\alpha = \sqrt{\frac{-\sec(2\theta)}{K}} \tag{27}$$

The transition from Equation 25 to Equation 26 is due to the double-angle formula. See Figure 4 for a visual demonstration that rescaling $\mathbf{v^0}$ by $\alpha$ works for the full range of $\theta$ values in one dimension. To extend this insight to arbitrary angles and curvatures, we define an $\alpha(\theta, K)$ function

$$\alpha(\theta, K) = \sqrt{\frac{-\sec(2\theta)}{K}} = \frac{\sqrt{-\sec(2\theta)}}{\sqrt{K}}. \tag{28}$$

Figure 4: Rescaling basis vector $\mathbf{v^0} = \langle \sin(\theta), \cos(\theta) \rangle$ by $\alpha(\theta, 1) = \sqrt{-\sec(2\theta)}$ produces a point on $\mathbb{H}^{1,1}$ for all $\theta$ values between $\pi/4$ and $3\pi/4$.

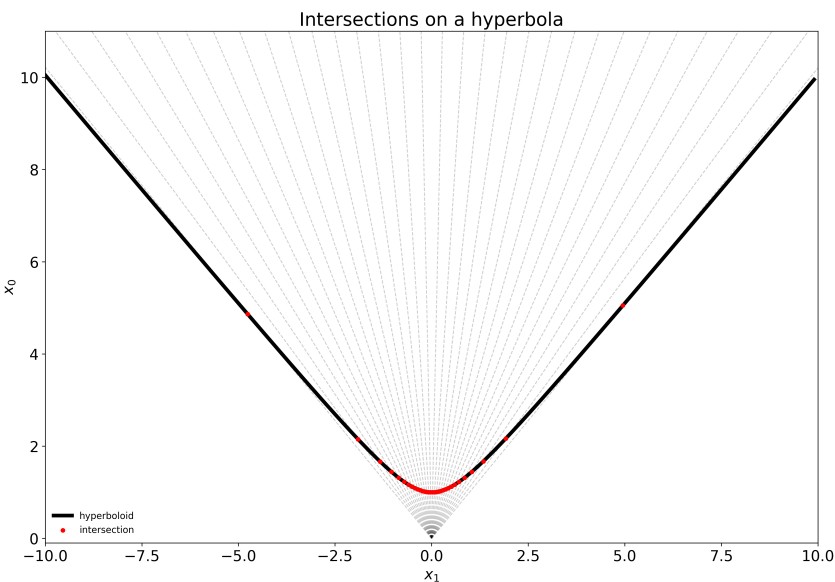

### A.2.3 GEODESIC ARCS

As shown in Chami et al. (2019), a geodesic arc in $\mathbb{H}^{D,K}$ can be characterized as $\cosh(t)\mathbf{v}^* + \sinh(t)\mathbf{u}^*$ for any pair of vectors $(\mathbf{u}^*, \mathbf{v}^*)$ where

$$\langle \mathbf{u}^*, \mathbf{u}^* \rangle_{\mathcal{L}} = 1/K \tag{29}$$
$$\langle \mathbf{v}^*, \mathbf{v}^* \rangle_{\mathcal{L}} = -1/K \tag{30}$$
$$\langle \mathbf{u}^*, \mathbf{v}^* \rangle_{\mathcal{L}} = 0. \tag{31}$$

All $\mathbf{u^d}$ vectors described in Equation 20, being purely spacelike, satisfy Equation 29 if rescaled by a factor of $\sqrt{K}$ so that their norms are $1/K$. We arbitrarily choose to use $\mathbf{u^2}/\sqrt{K}$ in our paramterization. Since $\alpha(\theta, K)\mathbf{v^0}$ lies on $\mathbb{H}^{D,K}$, it satisfies Equation 30. Since $\mathbf{v^0}$ and $\mathbf{u^d}$ are disjoint in their nonzero dimensions, they trivially satisfy Equation 31 for any scaling factors. Letting $t$ vary freely, the geodesic is given by

$$\mathbf{g^1}(\theta, K, t) = \cosh(t) \cdot \alpha(\theta, K) \cdot \mathbf{v^0} + \sinh(t)\mathbf{u^2}/\sqrt{K} \tag{32}$$

$$= \langle \cosh(t) \cdot \alpha(\theta, K) \cdot \sin(\theta), \ \cosh(t) \cdot \alpha(\theta, K) \cdot \cos(\theta), \ \sinh(t)/\sqrt{K}, \ 0, \ldots \rangle. \tag{33}$$

The full geodesic arc over all possible values of $t$ is given by

$$\mathbf{G^1}(\theta, K) = \left\{ \mathbf{g^1}(\theta, K, t) \ : \ t \in \mathbb{R} \right\}. \tag{34}$$

### A.2.4 GEODESIC SUBMANIFOLDS

For didactic purposes, we first extend the geodesic arc $\mathbf{G^1}$ to a 2-dimensional submanifold $\mathbf{G^2}$.

Since any point in $\mathbf{G^1}(\theta, K)$ is on $\mathbb{H}^{D,K}$, it has Minkowski norm $-1/K$ and therefore satisfies Condition 30.

By our convention, we use $\mathbf{u^d}$ vectors sequentially to construct geodesics. Therefore, if $\mathbf{G^d}$ is $d$-dimensional, then all $\mathbf{u^{d'}}$ for $d + 2 \le d' \le D$, being unused in the construction of the geodesic, remain orthogonal to all $\mathbf{v^d} \in \mathbf{G^d}$ and continue to satisfy Condition 31. In particular, $\mathbf{u^3}$ is the smallest unused $\mathbf{u^d}$ vector. Being spacelike, all $\mathbf{u^{d'}}$ continue to satisfy Condition 29.

For our next geodesic, we apply Equation 32 recursively to any $\mathbf{v^1} \in \mathbf{G^1}(\theta, K)$ and $\mathbf{u^3}/\sqrt{K}$:

$$\mathbf{g^2}(\theta, K, t, t') = \cosh(t')\mathbf{v^1} + \sinh(t')\mathbf{u^3}/\sqrt{K} \tag{35}$$

$$= \cosh(t')\mathbf{g^1}(\theta, K, t) + \sinh(t')\mathbf{u^3}/\sqrt{K} \tag{36}$$

$$= \cosh(t')(\cosh(t)\mathbf{v^0} + \sinh(t)\mathbf{u^2}) + \sinh(t')\mathbf{u^3}/\sqrt{K} \tag{37}$$

The geodesic submanifold created by intersecting the 3-plane with basis vectors $\{\mathbf{v^0}, \mathbf{u^2}, \mathbf{u^3}\}$ with $\mathbb{H}^{D,K}$ corresponds to the set of all values of $\mathbf{g^2}$ for $(t, t') \in \mathbb{R}^2$:

$$\mathbf{G^2}(\theta, K) = \{\mathbf{g^2}(\theta, K, t, t') \; : \; (t, t') \in \mathbb{R}^2\} \tag{38}$$

Using the remaining $\mathbf{u^d}$ vectors in ascending order from $d = 3$ to $D$, we can recursively parameterize the full geodesic submanifold resulting from intersecting $\mathbf{P}(\theta)$ with $\mathbb{H}^{D,K}$:

$$\mathbf{G^d}(\theta, K) = \{\sinh(t)\mathbf{u^{d+1}}/\sqrt{K} + \cosh(t)\mathbf{v^{d-1}} : \mathbf{v^{d-1}} \in \mathbf{G^{d-1}}(\theta, K), \; t \in \mathbb{R}\} \tag{39}$$

### A.2.5 VISUALIZATION-SPECIFIC DETAILS

**Visualization assumptions.** The first step to plotting a learned decision boundary is to find closed form equations for the intersection between the hyperboloid and the plane. To this end, we make a number of simplifying assumptions. First of all, we restrict ourselves to the hyperboloid $\mathbb{H}^{2,1}$. For decision tree visualization, hyperplanes can be inclined along dimensions 1 or 2; therefore, we cannot assume that our first dimension contains the split. Instead, we parameterize our plane as $\mathbf{P}(\theta, d)$. The details of geodesics are the same as Equation 34, but dimensions 1 and 2 may be exchanged when $d = 2$. We also assume that the plane actually intersects $\mathbb{H}^{2,1}$, meaning $\pi/4 < \theta < 3\pi/4$.

**Poincaré disk projection.** Since $\mathbb{H}^{2,1}$ is actually a 3-dimensional object for visualization purposes, it is easier to visualize as a point on the Poincaré disk $\mathbb{P}^{2,1}$. Thus, we convert coordinates in $\mathbb{H}^{2,1}$ to $\mathbb{P}^{2,1}$ using Equation 16. For geodesics, we sample 1,000 points uniformly from $(-10, 10)$ and convert these: this is sufficient to draw a smooth arc on $\mathbb{P}^{2,1}$.

**Subspace coloring.** For better visualizations, it is also necessary to partition $\mathbb{P}^{2,1}$ so that:

1. Decision boundaries are only rendered in the correct subtrees. For instance, if a boundary operates in the left subtree of a higher split, then it should only be drawn in the half of $\mathbb{P}^{2,1}$ where the left subtree is actually active.

2. The space is partitioned fully by the leaves of the decision tree, and can therefore be colored according to the majority class at each leaf node.

To do this, we recursively feed in a mask at each plotting iteration. This mask turns off plotting for inactive regions of the Poincaré disk. At the leaf level, every point on the Poincaré disk is active in only one mask, and therefore can be used to plot majority classes.

### A.3 MIDPOINT ANGLES

Now we consider how to find $\theta_m$, the midpoint between two angles $\theta_1$ and $\theta_2$. One option is to take the midpoint naively by taking the average of two angles:

$$\theta_{m,\text{naive}} = \frac{\theta_1 + \theta_2}{2}. \tag{40}$$

If we assume without loss of generality that $\sin(\theta_1) < \sin(\theta_2) < \pi/2$ (i.e. $\theta_2$ hits higher on the hyperboloid than $\theta_1$), then $\theta_{m,\text{naive}}$ will hit closer to $\theta_1$. Instead, we want some function $F(\theta_1, \theta_2) = \theta_m \in [\theta_1, \theta_2]$ such that $\delta(\theta_1, \theta_m) = \delta(\theta_2, \theta_m)$. To do this, we need to compute hyperbolic distances, so we use the pseudo-Euclidean metric in our ambient Minkowski space. In particular, we have the distance between two points defined as:

$$\delta(u, v) = \cosh^{-1}(-\langle u, v \rangle_{\mathcal{L}}) \tag{41}$$

$$= \cosh^{-1}(x_0 y_0 - x_d y_d) \tag{42}$$

$$= \ln\left(x_0 y_0 - x_d y_d + \sqrt{(x_0 y_0 - x_d y_d)^2 - 1}\right) \tag{43}$$

This assumes that all dimensions besides $0$ and $d$ are $0$, and fixes the point on the intersection between $\mathbb{H}^{D,K}$ and the decision hyperplane as the frame of reference for all distances. Using the definition of $\alpha(\theta, K)$ in Equation 28, we can simplify the conditions under which $\theta_m$ is an equidistant midpoint of $\theta_1$ and $\theta_2$:

$$\delta(\theta_a, \theta_b) := \cosh^{-1}\left(\alpha(\theta_a, K)\alpha(\theta_b, K)\cos(\theta_a + \theta_b)\right) \tag{44}$$

This distance function is quite nonlinear, as seen in Figure 5, which corroborates the inappropriateness of simply taking the mean between two angles as a midpoint. Instead, we set the distances $\delta(\theta_1, \theta_m)$ and $\delta(\theta_m, \theta_2)$ equal and simplify. For conciseness, we define the shorthand $\alpha_n := \alpha(\theta_n, K)$:

$$\cosh^{-1}(\alpha_1 \alpha_m \cos(\theta_1 + \theta_m)) = \cosh^{-1}(\alpha_m \alpha_2 \cos(\theta_m + \theta_2)) \tag{45}$$

$$\alpha_1 \alpha_m \cos(\theta_1 + \theta_m) = \alpha_m \alpha_2 \cos(\theta_m + \theta_2) \tag{46}$$

$$\alpha_1 \cos(\theta_1 + \theta_m) = \alpha_2 \cos(\theta_m + \theta_2) \tag{47}$$

$$\frac{\sqrt{-\sec(2\theta_1)}}{\sqrt{K}} \cos(\theta_1 + \theta_m) = \frac{\sqrt{-\sec(2\theta_2)}}{\sqrt{K}} \cos(\theta_m + \theta_2) \tag{48}$$

$$\sec(2\theta_1) \cos^2(\theta_1 + \theta_m) = \sec(2\theta_2) \cos^2(\theta_m + \theta_2) \tag{49}$$

$$\frac{\cos^2(\theta_1 + \theta_m)}{\cos(2\theta_1)} = \frac{\cos^2(\theta_m + \theta_2)}{\cos(2\theta_2)} \tag{50}$$

$$\cos(2\theta_2) \cos^2(\theta_1 + \theta_m) = \cos(2\theta_1) \cos^2(\theta_2 + \theta_m) \tag{51}$$

$$\cos(2\theta_2)(\cos(\theta_1)\cos(\theta_m) - \sin(\theta_1)\sin(\theta_m))^2 = \cos(2\theta_1)(\cos(\theta_2)\cos(\theta_m) - \sin(\theta_2)\sin(\theta_m))^2 \tag{52}$$

$$\cos(2\theta_2)(\cos(\theta_1)\cot(\theta_m) - \sin(\theta_1))^2 = \cos(2\theta_1)(\cos(\theta_2)\cot(\theta_m) - \sin(\theta_2))^2. \tag{53}$$

This is a quadratic equation in $\cot(\theta_m)$ expressed as $U\cot(\theta_m)^2 + W\cot(\theta_m) + U' = 0$, where:

$$U := \cos(2\theta_2)\cos^2(\theta_1) - \cos(2\theta_1)\cos^2(\theta_2)$$
$$= (2\cos^2(\theta_2) - 1)\cos^2(\theta_1) - (2\cos^2(\theta_1) - 1)\cos^2(\theta_2)$$
$$= \cos^2(\theta_2) - \cos^2(\theta_1) \tag{54}$$
$$W := \cos(2\theta_2) \cdot 2\cos(\theta_1)\sin(\theta_1) - \cos(2\theta_1) \cdot 2\cos(\theta_2)\sin(\theta_2)$$
$$= \cos(2\theta_2)\sin(2\theta_1) - \cos(2\theta_1)\sin(2\theta_2)$$
$$= \sin(2\theta_1 - 2\theta_2) \tag{55}$$
$$U' := \cos(2\theta_2)\sin^2(\theta_1) - \cos(2\theta_1)\sin^2(\theta_2)$$
$$= (1 - 2\sin^2(\theta_2))\sin^2(\theta_1) - (1 - 2\sin^2(\theta_1))\sin^2(\theta_2)$$
$$= \sin^2(\theta_1) - \sin^2(\theta_2)$$
$$= \cos^2(\theta_2) - \cos^2(\theta_1), \tag{56}$$

Since $U = U'$ we simplify further and solve $\cot^2(\theta_m) - 2V\cot(\theta_m) + 1 = 0$ where:

$$
\begin{aligned}
V := -W/2U &= \frac{-\sin(2\theta_1 - 2\theta_2)}{2(\cos^2(\theta_2) - \cos^2(\theta_1))}) \\
&= \frac{-\sin(2\theta_1 - 2\theta_2)}{2(\cos(\theta_2) + \cos(\theta_1))(\cos(\theta_2) - \cos(\theta_1))} \\
&= \frac{-\sin(2\theta_1 - 2\theta_2)}{-8\cos(\frac{\theta_1+\theta_2}{2})\cos(\frac{\theta_2-\theta_1}{2})\sin(\frac{\theta_1+\theta_2}{2})\sin(\frac{\theta_2-\theta_1}{2})} \\
&= \frac{\sin(2\theta_1 - 2\theta_2)}{2\sin(\theta_2 + \theta_1)\sin(\theta_2 - \theta_1)}
\end{aligned}
\tag{57}
$$

The solutions for the quadratic equation are $\cot(\theta_m) = V \pm \sqrt{V^2 - 1}$; specifically, we have

$$
\theta_m = \begin{cases}
\theta_1 & \text{if } \theta_1 = \theta_2 \\
\cot^{-1}\left(V - \sqrt{V^2-1}\right) & \text{if } \theta_1 < \pi - \theta_2 \\
\cot^{-1}\left(V + \sqrt{V^2-1}\right) & \text{if } \theta_1 > \pi - \theta_2
\end{cases}
\tag{58}
$$

Figure 5: A plot of function $\delta(\pi/4 + .01, \theta)$ as $\theta$ varies from $\pi/4$ to $3\pi/4$. This plot reveals the nonlinearity of the angle distance function.

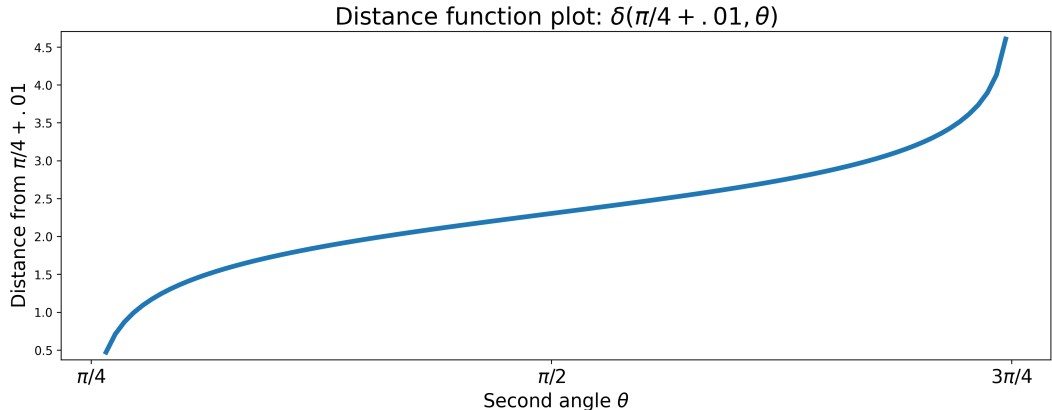

### A.4   MIXTURE OF GAUSSIANS ON HYPERBOLIC MANIFOLDS

We modify the method put forward in Nagano et al. (2019) to sample Gaussians in hyperbolic space. We briefly reiterate their method to sample a single Gaussian in $\mathbb{H}^{D,K}$:

1. Choose a point $\mu$ in $\mathbb{H}^{D,K}$ to be the mean of your Gaussian.

2. Sample $\mathbf{X}$ as $n$ samples from a Euclidean multivariate Gaussian with mean $0$ and any covariance $\boldsymbol{\Sigma}$ in $D$ dimensions.

3. Transform $\mathbf{X}$ into $\mathbf{X}'$, a set of vectors in $T_0\mathbb{H}^{D,K}$ (the tangent plane of $\mathbb{H}^{D,K}$ at the origin), by appending 0 in the timelike dimension.

4. Use parallel transport from the origin to $\mu$, turning $\mathbf{X}'$ into $\mathbf{X}''$, a set of vectors in $T_\mu\mathbb{H}^{D,K}$.

5. Use the exponential map at $\mu$ to map $\mathbf{X}''$ to $\mathbf{X}'''$, a set of points on the surface of $\mathbb{H}^{D,K}$.

For each Gaussian in our mixture, we perform this exact procedure. We choose our set of $n$ Gaussian means by sampling $n$ vectors from $\mathcal{N}(0, \mathbf{I})$ in the tangent plane and then exponentially mapping them directly to $\mathbb{H}^{D,K}$; in other words, we follow the above procedure but skip step (4) because $\mu$ is not defined yet (or, equivalently, because $\mu$ is the origin).

Additionally, each covariance matrix is generated by drawing $D$ $D$-dimensional samples, $\mathbf{C} \sim \mathcal{N}(0, \mathbf{I})$, and then letting $\mathbf{\Sigma} = \mathbf{C}\mathbf{C}^T$. The entire covariance matrix is optionally rescaled by a user-set noise scalar $a$ and divided by $D$. That is,

$$\mathbf{\Sigma} = a\frac{\mathbf{C}\mathbf{C}^T}{D}. \tag{59}$$

This procedure is repeated $n$ times to yield $n$ distinct covariance matrices.

Finally, class probabilities are determined by drawing $n$ values from $U(0, 1)$ and normalizing them by their sum. Each point in a sample is assigned a class that determines its $\mu$ and $\mathbf{\Sigma}$. We implement this method using the geomstats package in Python (Miolane et al., 2018), which supports vectorized versions of parallel transport and exponential maps with differing destinations.

## A.5   EQUIVALENCE OF MINKOWSKI AND EUCLIDEAN DOT-PRODUCTS

Most papers on hyperbolic geometry use Minkowski products. For instance, implementing the support vector machine objective in Cho et al. (2018) relies on Minkowski products, which are crucial for the optimization procedure they specify.

In our case, it is sufficient to use the more intuitive Euclidean formulation, even though, in practice, Minkowski space is not equipped with Euclidean products. Intuitively, Euclidean inner products (dot-products) accurately capture whether a point is to one side of a plane or another, which is all that is needed for a decision tree classifier. However, we show further that Euclidean dot-products for HYPERDT decision boundaries have an interpretation in terms of Minkowski products:

The sparse Euclidean dot-product we determined in Equation 9 is:

$$S(x) = \text{sign}\left(\max\left(0, \ (\sin(\theta)x_d - \cos(\theta)x_0)\right)\right) \tag{60}$$

And, since the Minkowski inner product is simply the Euclidean dot-product with the sign of the timelike dimension flipped, we can equivalently say

$$S(x)_{\text{Minkowski}} = \text{sign}\left(\max\left(0, \ (\sin(\theta)x_d + \cos(\theta)x_0)\right)\right) \tag{61}$$

Any $\theta$ in the Euclidean case, if substituted for $-\theta$ in the Minkowski case, will yield the same $\cos(\theta)$ and a negated $\sin(\theta)$. That is,

$$\sin(\theta)x_d - \cos(\theta)x_0 = a \tag{62}$$
$$\sin(-\theta)x_d + \cos(-\theta)x_0 = b \tag{63}$$
$$-\sin(\theta)x_d + \cos(\theta)x_0 = b \tag{64}$$
$$\sin(\theta)x_d - \cos(\theta)x_0 = -b \tag{65}$$
$$a = -b \tag{66}$$

That is, for any angle $\theta$ yielding a particular split $S$ over a dataset $\mathbf{X}$, evaluating the split using Minkowski inner products with the angle $-\theta$ produces an equivalent split. The sets are exactly the same, but the sign of the dot-product is flipped.

## A.6   ADDITIONAL EXPERIMENTS

### A.6.1   SCALING

In Figure 3, we showed that HYPERDT runtime scales linearly with sample size—an improvement over the exponential scaling of HORORF. In this section, we show how runtime scales with the number of data points, number of dimensions, maximum depth of decision trees, and total number of estimators. Since we are not comparing to slower methods, we can extend our analysis to substantially more than the upper limit of 800 samples we use in the main section of the paper.

**Procedure.**   Unless noted otherwise, we test the runtime and F1-micro accuracy of HYPERRF with a maximum depth of 3 and 12 trees, consistent with the HYPERRF results in Table 1. We restrict ourselves to Gaussian mixtures of five classes rather than two, since this more challenging classification task has a greater range of F1-micro scores. Unless noted otherwise, we generated 1,000 points for 20 distinct trials, without cross-validation and using a test set size of 200 points.

We carried out four distinct scaling experiments, recording runtime and accuracy as we varied:

1. The number of points generated from 100 to 3,000

2. The number of dimensions from 2 to 64

3. The total number of decision trees in a forest from 1 to 30

4. The max depth of each decision tree from 1 to 20

Additionally, for the final max depth experiment, we also tested SCIKIT-LEARN Euclidean random forests and HORORF, also using 12 predictors. For this portion of the experiment, we restrict ourselves to 800 samples from a 2-dimensional, 2-class mixture of Gaussians.

**Results.** Times and F1-micro scores for the four HYPERRF scaling experiments are shown in Figure 6. This figure shows that runtime scales linearly with the number of samples, dimensions, and trees, with little effect on overall prediction accuracy. Interestingly, the runtime levels off rather than growing exponentially for the maximum depth experiment as one might expect given the $2^{max\_depth}$ splits the predictor is allowed to make. This is because the actually achieved depth tops out when the training set is perfectly divided into homogeneous subregions of the decision space, and further splits are not made. Additionally, F1-micro scores decline slightly with increasing tree depth, likely due to overfitting.

Since maximum depth is the only parameter with a particularly interesting relationship to prediction accuracy, we explored it further in the context of the other predictors evaluated in the paper. In Figure 7, we compare the F1-micro scores of HYPERRF, HORORF, and Euclidean random forests, and find that HYPERRF has a consistent advantage over other predictors at the same maximum depth; however, as maximum depth increases, this advantage becomes less prominent. This result speaks both to the general ability of elaborate random forests to model data from arbitrary probability distributions, and to the parsimony of HORODT-based methods in modeling hierarchical data.

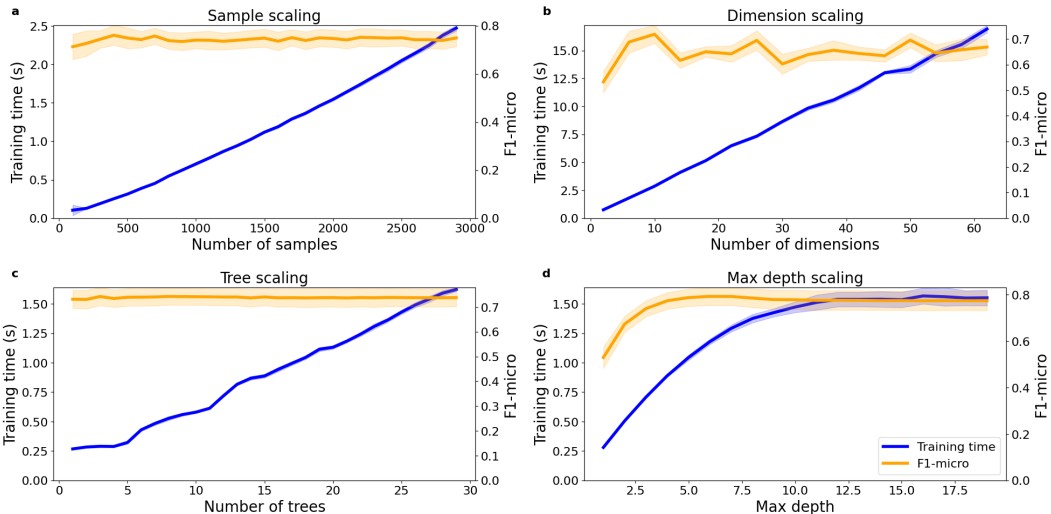

Figure 6: Observed runtimes when varying: (a) number of samples, (b) dimensionality, (c) number of trees, and (d) maximum depth in a simulated Gaussian mixture classification problem. We observe linear scaling for (a), (b), and (c), and possibly sublinear scaling with maximum tree depth. Shaded regions represent 95% confidence intervals.

### A.6.2 COMPARISON TO OTHER HYPERBOLIC CLASSIFIERS

In the main body of the paper, we restricted ourselves to classifiers based on decision trees and random forests. However, there are a number of other capable and powerful classifiers for use on hyperbolic data that warrant evaluation. In this section, we evaluate hyperbolic support vector machines and logistic regression classifiers against some of our benchmarks.

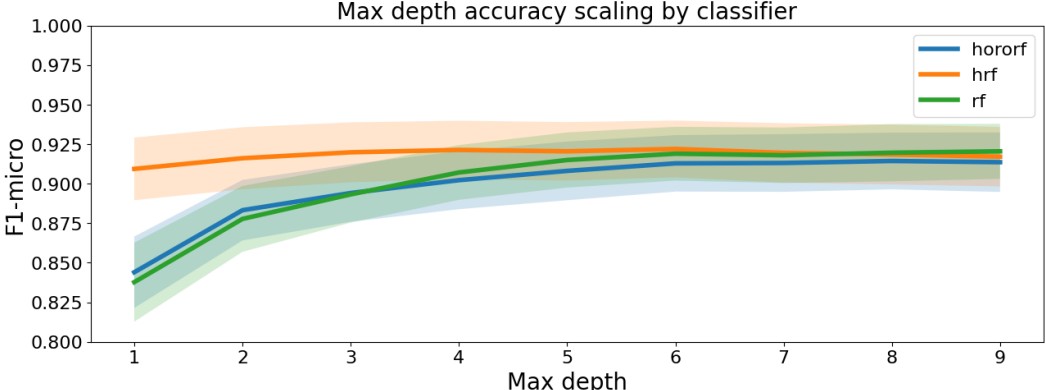

Figure 7: F1-micro scores on a Gaussian mixture classification problem with 2 classes, 2 dimensions, and 800 samples. Shaded regions represent 95% confidence intervals.

**Procedure.** We used the implementation of hyperbolic support vector machines provided by Agibetov et al. (2019)[1], and the implementation of hyperbolic logistic regression provided by Bdeir et al. (2023)[2]. For each trial, we generated 800 points from a gaussian mixture with 2 classes and evaluated it using the F1-micro score under 5-fold cross validation. We did this for 10 trials total, in 2, 4, 8, and 16 dimensions.

In addition to the hyperbolic support vector machine and logistic regression classifiers, we evaluated their Euclidean counterparts, which are implemented in SCIKIT-LEARN Pedregosa et al. (2011). We only benchmark against HYPERDT, which is simpler and slightly less accurate than HYPERRF, for fairness.

**Results.** We report average F1-micro scores for each classifier in each dimension in Table 2. Following the conventions of the paper, we additionally mark statistically significant differences from HYPERDT with an asteristk. In total, HYPERDT was a statistically significant improvement over each classifier in at least one dimension, and was the best classifier in half the cases. This shows a consistent advantage over the other classifiers, which we can expect to be further improved by use of HYPERRF.

| $D$ | Logistic Regression | Hyperbolic Logistic Regression | Hyperbolic Support Vector Classifier | Support Vector Classifier | HYPERDT |
|---|---|---|---|---|---|
| 2 | 90.11* | 88.65* | 81.50* | 80.96* | **91.88** |
| 4 | 99.20 | **99.41** | 96.94* | 85.56* | 99.30 |
| 8 | 99.97 | 99.96 | **100.00** | 79.06* | 99.96 |
| 16 | 99.99 | **100.00** | 98.12* | 86.99* | **100.00** |

Table 2: Micro-averaged F1 scores under 5-fold cross validation averaged over 10 seeds for each classifier and dimension. Bold indicates the best score for that dimension; asterisks indicate a statistically significant difference from HYPERDT as determined by a paired $t$-test.

### A.6.3 RANDOM FORESTS IN OTHER GEOMETRIES

While no representation of hyperbolic space is explicitly compatible with axis-aligned splits, it is worth exploring the possibility that other representations nonetheless lend themselves better to treatment by decision tree or random forest classifiers than the hyperboloid model does; more specifically, it is worth testing HYPERDT and HYPERRF against a greater range of embeddings to ensure we are making a fair comparison.

---

[1] https://github.com/plumdeq/hsvm
[2] https://github.com/danielbinschmid/HyperbolicCV/tree/main

**Procedure.** Analogously to other sections, we restrict ourselves to 800 samples from 2-class mixtures of Gaussians in 2, 4, 8, and 16 dimensions. We record F1-micro scores under 5-fold cross-validation, averaged over 10 seeds. In this case, we evaluated HYPERDT, HYPERRF, and scikit-learn implementations of Euclidean decision trees and random forests for each sample.

We converted each sample to Euclidean, Hyperboloid, Klein disk, and Poincaré disk coordinates. To get Euclidean coordinates, we applied the logarithmic map to project points from $\mathbb{H}^{D,K}$ to the tangent plane at the origin.

**Results.** Table 3 shows a comparison of HYPERDT and HYPERRF to each of these geometries. Both HYPERDT and HYPERRF show substantial, statistically significant advantages over their Euclidean counterparts when applied to the hyperboloid model, and this is the only consistent trend in the data. Only the Poincaré disk in two dimensions beat HYPERRF with statistical significance.

Interestingly, the Klein disk embeddings performed well (without statistical significance) in two and three dimensionalities for decision trees and random forests, respectively. This is likely because geodesics in the Klein model are represented with straight lines, so the axis-parallel splits used by Euclidean decision tree algorithms are also geodesic decision boundaries. This is yet another point in support of geodesic decision boundaries yielding improved classification performance.

| Model | Geometry | Dimensions | | | |
| | | 2 | 4 | 8 | 16 |
| --- | --- | --- | --- | --- | --- |
| Decision Tree | Euclidean | 91.86 | 99.15 | 99.94 | 99.97 |
| | Hyperboloid | 90.16* | 98.34* | 99.91 | 99.99 |
| | Klein | **91.89** | 99.29 | **99.96** | 99.99 |
| | Poincaré | 91.85 | **99.30** | **99.96** | **100.00** |
| HYPERDT | Hyperboloid | 91.88 | **99.30** | **99.96** | **100.00** |
| HYPERRF | Hyperboloid | 91.91 | 99.42 | 99.96 | **100.00** |
| Random Forest | Euclidean | 92.09 | 99.28 | 99.97 | **100.00** |
| | Hyperboloid | 89.80* | 98.40* | 99.95 | **100.00** |
| | Klein | 92.05 | **99.46** | **100.00** | **100.00** |
| | Poincaré | **92.26*** | 99.45 | 99.97 | **100.00** |

Table 3: F1-micro scores for Euclidean and hyperbolic random forests and decision trees when applied to a variety of hyperbolic coordinate systems. Bolded scores are the best for that dimension; asterisks represent statistical significance.

### A.6.4 HYPERBOLIC IMAGE EMBEDDINGS

CLIP is a contrastive deep learning model with a joint text-image embedding space (Radford et al., 2021). Recently, Desai et al. (2023) developed MERU, a modified version of CLIP which encodes images and text into $\mathbb{H}^{512,0.1}$. They impose a hierarchical structure on image an text embeddings: since images are more specific than the sentences which describe them, they encourage image embeddings to have larger values in dimension 0 than their corresponding text embeddings. In this case, the root of the hyperboloid represents the most general possible concept. Learning this hierarchy, MERU matches or outperforms CLIP on zero-shot text-to-image and image-to-text *retrieval* tasks. However, it matches or underperforms CLIP on zero-shot image *classification* tasks.

**Procedure.** We hypothesized that MERU classification performance suffers because Desai et al. (2023) used a logistic regression classifier, designed for Euclidean, not hyperbolic, space. To test this, we perform zero-shot image classification on CIFAR-10 on pretrained ViT S/16 MERU and CLIP image embeddings using Euclidean and hyperbolic random forests.[3] All forests used 10 estimators with a maximum depth of 5.

We also experiment with partial and full image embeddings. CLIP passes an image throught an image-encoder, a linear projection layer, and then an $L_2$ normalization layer (project to the unit hypersphere). Similarly, MERU passes an image through an image-encoder, a linear projection

---

[3]Pretrained models at `https://github.com/facebookresearch/meru`.

| Embedding | Predictor | CLIP Accuracy | MERU Accuracy |
|-----------|-----------|---------------|---------------|
| Encoder | Baseline | 89.60 | 89.70 |
| Encoder | Logistic regression | 87.60 | **90.15** |
| Encoder | Random forest | 82.00 | 86.05 |
| Encoder + LP | Random forest | 84.20 | 84.85 |
| Encoder + LP + map | Random forest | 84.00 | 85.10 |
| Encoder + LP + map | HYPERRF | — | 86.20 |

Table 4: Per-class average accuracies on zero-shot CIFAR-10 classification benchmarks. HYPERRF and SCIKIT-LEARN benchmarks are new, with baseline accuracies taken from Table 7 of Desai et al. (2023). *LP* stands for linear projection. Note that *map* means $L_2$ normalization for CLIP and exponential map to the hyperboloid for MERU. HYPERRF can only be evaluated on MERU with linear projection and exponential map applied, since all other representations are Euclidean.

layer, and then an exponential map (project to the hyperboloid). In their experiments, Desai et al. (2023) performed classification only on the image-encoder outputs, which lie in Euclidean space for both CLIP and MERU. However, this approach ignores (1) the information provided by the projection layer and cannot leverage (2) the hierarchical structure gained from projecting to the hyperboloid. We thus exeriment with embeddings with different combinations of layers.

**Results.** The results of this experiment are summarized in Table 4, alongside the reported accuracies from the original paper. We do not report standard deviation because we use the same train/test split as Desai et al. (2023).

First, we find that Euclidean random forests perform better on MERU-encoded data than CLIP-encoded data. This suggests that, in a linear probing context, MERU representations actually are more separable with respect to CIFAR-10 classes.

Additionally, we show that hyperboloid random forests on MERU encodings with linear projection and exponential mapping to the hyperboloid outperform all other combinations of classifiers and embeddings. These findings both demonstrate that hyperbolic embeddings for image-text joint embeddings enhance zero-shot classification performance and that hyperbolic random forests outperform their Euclidean counterparts in this embedding space.

HYPERRF outperforms all Euclidean random forests, suggesting it is the best forest-type classifier for this task. However, we fail to beat the benchmark value reported in Desai et al. (2023) for logistic regression. We reproduce similar accuracies on our own implementation of logistic regression.

### A.6.5 WORDNET EMBEDDINGS

Hyperbolic embeddings of Wordnet Fellbaum (2010) are another popular benchmark for hyperbolic classifiers. We extend our analysis to WordNet classification tasks.

**Procedure.** We use the WordNet embeddings and labels provided in the Github repository for Doorenbos et al. (2023)[4]. These are split into binary classification tasks, where embeddings are labeled according to whether or not they belong to a certain class of things (animal, group, mammal, and so on), and multiclass classification tasks. For speed, we downsample the WordNet embeddings to 1,000 randomly-sampled points without rebalancing the classes. The exact same sample is seen by all classifiers. Each classifier is evaluated across 10 seeds using 5-fold cross-validation.

**Results.** The results for the WordNet experiment are shown in Table 5. These results continue to be very favorable for HYPERDT and HYPERRF: HYPERDT beats Sklearn and HORORF 8 times each, and HYPERRFbeats Sklearn 6 times and HORORF 8 times, all with statistical significance. No other models achieved a statistically significant advantage against any other models.

---

[4]https://github.com/LarsDoorenbos/HoroRF

| Data | | Decision Trees | | | Random Forests | | |
|---|---|---|---|---|---|---|---|
| | | HYPERDT | SCIKIT-LEARN | HORORF | HYPERRF | SCIKIT-LEARN | HORORF |
| Binary | Animal | **98.88**$^\dagger$ | 98.69 | 96.02 | **98.97**$^{\ddagger\dagger}$ | 98.16 | 96.22 |
| | Group | **94.65**$^{\ddagger\dagger}$ | 94.06 | 91.77 | **94.64**$^\dagger$ | 94.23 | 92.34 |
| | Mammal | **99.86**$^{\ddagger\dagger}$ | 99.33 | 98.92 | **99.87**$^{\ddagger\dagger}$ | 99.19 | 98.92 |
| | Occupation | 99.58 | 99.49 | **99.64** | 99.61 | 99.66 | **99.69** |
| | Rodent | **99.83** | 99.78 | 99.79 | 99.81 | **99.85** | **99.85** |
| | Solid | **99.11**$^{\ddagger\dagger}$ | 98.72 | 98.55 | **99.13**$^{\ddagger\dagger}$ | 98.50 | 98.45 |
| | Tree | **98.90**$^{\ddagger\dagger}$ | 98.59 | 98.46 | **99.01**$^{\ddagger\dagger}$ | 98.68 | 98.63 |
| | Worker | **98.69**$^\ddagger$ | 98.36 | 98.57 | **98.73** | 98.58 | 98.57 |
| Multi | Same level | **98.10**$^{\ddagger\dagger}$ | 97.33 | 96.98 | **98.31**$^{\ddagger\dagger}$ | 96.91 | 96.71 |
| | Nested | **89.44**$^{\ddagger\dagger}$ | 87.74 | 77.19 | **89.72**$^\dagger$ | 89.22 | 86.36 |
| | Both | **96.38**$^{\ddagger\dagger}$ | 95.60 | 91.22 | **96.67**$^{\ddagger\dagger}$ | 94.33 | 91.13 |

Table 5: Mean micro-F1 scores for classification benchmarks over 10 seeds and 5 folds. The highest-scoring decision tree and random forests are bolded separately. $*$ means a predictor beat HYPERRF, $\dagger$ means a predictor beat HORORF, and $\ddagger$ means a predictor beat SCIKIT-LEARN, with $p < 0.05$.

### A.6.6 MIDPOINT ABLATION

Because the use of the midpoint formula laid out in Equation 10 was guided by intuition rather than actual performance, a worthwhile experiment is to check the effect that substituting this operation with the naive midpoint calculation has on the accuracy and efficiency of our algorithm.

**Procedure.** We test the effect of substituting the geodesic midpoint calculations with naive midpoint calculations across 10 seeds and a single train-test split. We check this for 100, 200, 400, and 800 points and 2, 4, 8, and 16 dimensions, recording total runtime and F1-micro score on the held-out test set (20% of the sample).

**Results.** Figure 8 shows the results of the midpoint ablation study. Across almost all sample sizes and dimensions, substituting the naive midpoint computation resulted in a marked reduction in accuracy—although with the benefit of a slight reduction in runtime. We believe that not only is this accuracy-performance tradeoff favorable to the more complicated midpoint computation, it is also theoretically more justified (all things being equal, you should prefer to place the decision boundary exactly in between two differing classes). Thus, we elect to keep the midpoint computation as it is.

### A.7 STATISTICAL TESTING

For the benchmarks reported in Table 1, we provide statistical significance annotations. To test statistical significance, we used a two-tailed paired $t$-test comparing $F_1$ scores over 10 seeds and 5 folds. Each combination of dataset, dimension, and sample size was tested separately. We used a threshold of $p = 0.05$ on this test to determine statistical significance, and assigned the significance annotation to the predictor with the higher mean. We report full $p$-values in Table 6.

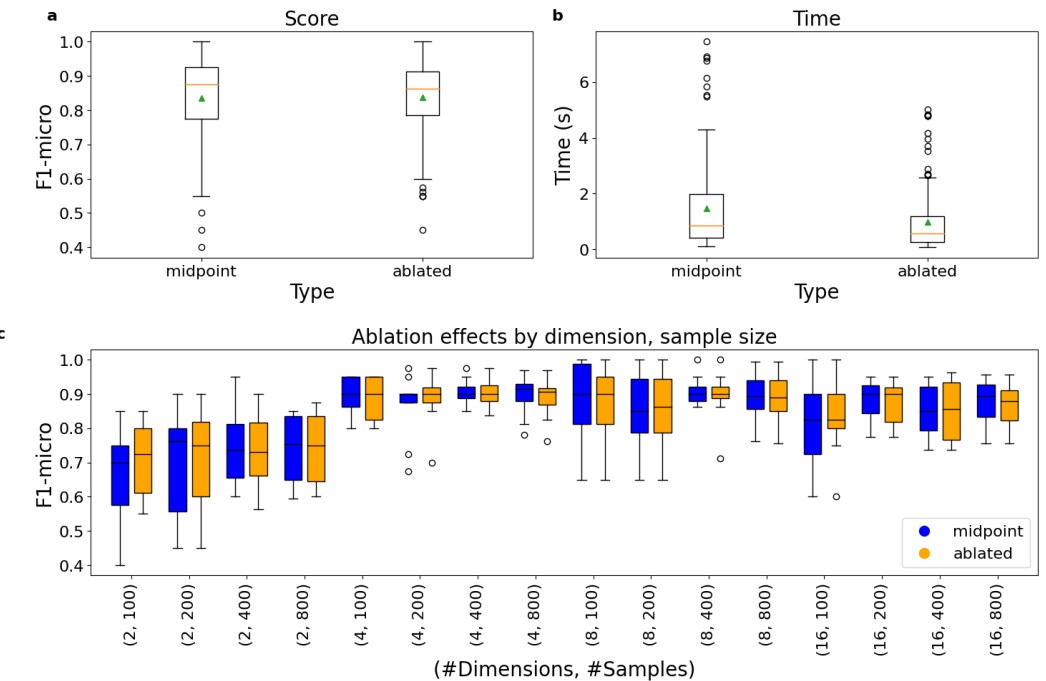

Figure 8: Ablation results for midpoint angle computations.

| Dataset | $D$ | $n$ | DT vs HYPERDT | DT vs HoroRF | HoroRF vs HYPERDT | RF vs HYPERRF | RF vs HoroRF | HoroRF vs HYPERRF |
|---|---|---|---|---|---|---|---|---|
| Gaussian | 2 | 100 | .204 | 4.56e-03 | .027 | .368 | 3.92e-03 | 1.16e-03 |
| | | 200 | .545 | .053 | 5.88e-04 | .960 | 1.39e-04 | .092 |
| | | 400 | 4.75e-03 | 4.86e-03 | 6.10e-05 | .867 | 6.71e-07 | 4.41e-05 |
| | | 800 | 1.52e-03 | 3.02e-04 | 1.05e-06 | .860 | 1.14e-10 | 2.55e-07 |
| | 4 | 100 | .067 | .358 | 8.29e-05 | 1.00 | 9.91e-06 | .417 |
| | | 200 | .036 | 6.50e-03 | 3.98e-05 | .705 | 1.58e-07 | .025 |
| | | 400 | 7.02e-03 | 5.43e-03 | 5.95e-03 | .514 | 7.62e-07 | 4.90e-03 |
| | | 800 | 4.29e-03 | 6.03e-04 | .034 | .062 | 5.19e-05 | 2.80e-03 |
| | 8 | 100 | .709 | .420 | 2.17e-03 | .252 | .233 | .083 |
| | | 200 | .821 | .766 | 1.14e-03 | .766 | 9.18e-04 | 1.00 |
| | | 400 | .785 | .569 | .012 | .533 | .011 | 1.00 |
| | | 800 | .133 | .103 | 2.41e-03 | .598 | 3.83e-04 | .598 |
| | 16 | 100 | .261 | .420 | .146 | 1.00 | .032 | .420 |
| | | 200 | .322 | .569 | .011 | .261 | .028 | .322 |
| | | 400 | .322 | — | .182 | .159 | .044 | .159 |
| | | 800 | — | — | .033 | .096 | — | — |
| NeuroSEED | 2 | 100 | .236 | .160 | .014 | .577 | 2.37e-03 | .824 |
| | | 200 | .037 | 8.32e-03 | 1.14e-05 | .138 | 1.54e-06 | .912 |
| | | 400 | .010 | 8.82e-06 | 6.84e-10 | .894 | 9.78e-11 | .048 |
| | | 800 | .878 | 1.25e-08 | 1.77e-10 | .807 | 2.86e-10 | 4.56e-03 |
| | 4 | 100 | .709 | .055 | 3.45e-17 | 3.34e-05 | 3.30e-17 | 2.02e-07 |
| | | 200 | .322 | 7.99e-05 | 2.57e-24 | 1.61e-11 | 3.05e-24 | 9.19e-15 |
| | | 400 | .159 | 2.75e-05 | 2.53e-24 | 6.10e-21 | 2.86e-24 | 6.94e-23 |
| | | 800 | .322 | 2.00e-10 | 1.12e-26 | 5.07e-20 | 1.22e-26 | 2.71e-24 |
| | 8 | 100 | .533 | 7.92e-03 | 2.65e-14 | 3.44e-10 | 2.47e-14 | 1.60e-05 |
| | | 200 | .622 | 6.27e-04 | 3.62e-20 | 8.01e-20 | 1.82e-19 | 4.50e-15 |
| | | 400 | .229 | 5.28e-08 | 4.62e-26 | 6.56e-32 | 8.02e-26 | 4.90e-27 |
| | | 800 | 1.00 | 6.99e-13 | 7.88e-37 | 9.02e-36 | 6.65e-37 | 1.65e-33 |
| | 16 | 100 | .322 | .595 | 2.98e-07 | .061 | 3.40e-08 | .123 |
| | | 200 | .279 | .880 | 4.98e-07 | .213 | 2.02e-07 | .180 |
| | | 400 | .569 | 2.00e-03 | 6.56e-08 | .120 | 4.82e-08 | 1.47e-04 |
| | | 800 | .659 | 1.26e-03 | 1.40e-14 | .114 | 1.62e-14 | 2.57e-07 |
| Polblogs | 2 | 979 | .118 | .571 | 1.23e-05 | 2.38e-05 | 9.15e-07 | 4.50e-06 |
| | 4 | 979 | .160 | .696 | 2.05e-07 | 1.68e-03 | 6.26e-08 | 1.63e-03 |
| | 8 | 979 | .047 | .301 | 1.11e-12 | 2.48e-10 | 1.87e-11 | 6.43e-10 |
| | 16 | 979 | .987 | .302 | 2.34e-10 | 3.64e-09 | 8.29e-11 | 1.65e-07 |

Table 6: Paired $t$-test values for benchmarks over 5 folds and 10 seeds. Missing values indicate that two sets of $F_1$ scores were identical. Statistically significant values are used to generate cell annotations in Table 1.

