# OpenReview forum: "Fast Hyperboloid Decision Tree Algorithms"
_ICLR.cc/2024/Conference — ICLR 2024 poster_

### Official Review · Reviewer_uhLf · 2023-10-24

**Soundness:** 3 good
**Presentation:** 3 good
**Contribution:** 2 fair
**Rating:** 5
**Confidence:** 4

**Summary:**

The paper introduces a novel extension of the Euclidean CART algorithm into the hyperbolic space, aptly named HyperDT. This extension involves a transformative shift from axis-aligned decision planes to geodesic submanifolds and replaces candidate thresholds with equidistance mid-angles. Notably, other components of HyperDT, including objective functions, remain consistent with the traditional CART framework. This adaptation allows HyperDT to achieve a similar asymptotic complexity as CART. Furthermore, the authors present another contribution in the form of HyperRF, a random forest model built upon the foundations of HyperDT. The paper showcases the good performance of these methods across a spectrum of datasets, including synthetic datasets like Gaussian mixtures, as well as real-world data sources such as biological sequences and graph embeddings. These empirical results underscore the effectiveness and versatility of the proposed techniques.

**Strengths:**

1. The utilization of a CART-like structure in the proposed methods greatly enhances efficiency, a feature that has been substantiated through a comparison with previous works like HoroRF.

2. The paper excels in presenting a lucid and methodical derivation of the proposed techniques. It ingeniously adapts Euclidean decision tree algorithms to the hyperbolic space by leveraging inner products. Additionally, the paper offers insightful closed-form equations for the decision boundaries and provides an in-depth explanation of the candidate hyperplane selection process.

3. Another notable aspect of the paper is its commitment to practicality. The authors have thoughtfully offered a Python implementation of the methods, aligning with the conventions of the scikit-learn API. This effort to bridge theory and practical application enhances the paper's value and accessibility to the research community.

**Weaknesses:**

My primary concerns are related to the CART-like design employed in the paper. As CART is known for its top-down, greedy approach, it tends to yield suboptimal solutions [1]. It is essential to acknowledge that the proposed method may also encounter this suboptimal behavior due to its similarity to CART. For additional concerns, please refer to the questions section.

[1] Mazumder, Rahul, Xiang Meng, and Haoyue Wang. "Quant-BnB: A Scalable Branch-and-Bound Method for Optimal Decision Trees with Continuous Features." *International Conference on Machine Learning*. PMLR, 2022.

**Questions:**

1. In Section 3.2, the authors mentioned that the interpretation of axis-aligned hyperplanes within the hyperboloid model is unclear. Can the authors provide a more detailed explanation of why the use of axis-aligned hyperplanes lacks clarity in this context? I am considering that any geodesic could potentially be represented by two axis-aligned or oblique planes, making the use of axis-aligned planes reasonable.

2. Additionally, does the geodesic boundary offer advantages beyond classification scores and interpretation? One potential advantage I am considering is that geodesic trees may require fewer nodes compared to axis-aligned trees to achieve similar classification scores. Could the authors include metrics, such as the average number of nodes used in the experiments presented in Table 1, to support this claim?

3. For a more comprehensive evaluation, could the authors include additional comparison results? As mentioned in the weaknesses section, there are now several global optimization methods, such as [1], that have demonstrated superior performance compared to CART. It would be valuable to include a comparison with at least one of these methods to illustrate the superior performance of geodesic decision boundaries over axis-aligned boundaries.

4. Ablation experiments: The paper lacks ablation studies or experiments to justify the design choices made in the proposed methods, including the selection of candidate hyperplanes. Since the equidistance mid-angles in the proposed method do not have closed-form solutions and rely on numerical solvers, there may be concerns about the algorithm's efficiency. It would be informative to compare the performance and execution times of the equidistance mid-angles with those of naive mid-angles to assess the impact of these choices on the methods' performance and behavior.

---

> ### Author Response · Authors · 2023-11-22
> **Thank you for your feedback!**
>
> We thank you for your comments on the efficiency, ingenuity, and practicality of our method.
>
> We have modified the paper to strengthen our conceptual contributions in response to your astute observations. Specifically, we have added a discussion of theoretical reasons to prefer geodesic boundaries and also added a discussion of more advanced decision tree algorithms, noting that the suboptimality of CART remains a concern in our model. We additionally added experiments showing HyperDT is more parsimonious than other decision trees (Section A.6.1) and added comparisons to other classifiers (A.6.2), other models of hyperbolic space (A.6.3), and a new WordNet classification benchmark (A.6.5). Although we have replaced the numerical approximation for midpoint angles with an analytic solution, we nonetheless carried out a midpoint ablation study in A.6.6. We now respond to your comments inline:
>
> __My primary concerns are related to the CART-like design employed in the paper. As CART is known for its top-down, greedy approach, it tends to yield suboptimal solutions [1]. It is essential to acknowledge that the proposed method may also encounter this suboptimal behavior due to its similarity to CART. For additional concerns, please refer to the questions section.__
>
> While the duration of the reply period limited us to adding comparisons against hyperbolic logistic regression and hyperbolic SVMs, HyperDT can be adapted to other decision tree algorithms besides CART. The decision to adapt CART was motivated by its widespread popularity, effectiveness in machine learning, and the desire to avoid confounding the comparison of predictors’ geometries by differences in optimization procedures. We have updated the related work section and conclusion to include a discussion of other decision tree frameworks and leave additional extensions for future work. We believe that, since Quant-BnB is based on axis-parallel splits, it is in principle possible to wed globally optimal optimization procedures with geodesic decision boundaries.
>
> __In Section 3.2, the authors mentioned that the interpretation of axis-aligned hyperplanes within the hyperboloid model is unclear. Can the authors provide a more detailed explanation of why the use of axis-aligned hyperplanes lacks clarity in this context? I am considering that any geodesic could potentially be represented by two axis-aligned or oblique planes, making the use of axis-aligned planes reasonable.__
>
> We have revised the discussion of geodesic decision boundaries in Sections 1.2, 3.2, and 5 to emphasize that, of the three classes of decision tree algorithms considered, ours is the only one that splits the space into topologically continuous subspaces closed under geodesic combinations—a property of axis-aligned decision splits one may take for granted in a proper Euclidean context. Additionally, we observe that, although it is possible to represent geodesics using a combination of planes, CART-based algorithms fit single hyperplanes at a time, creating hyperbolas. Specifically, “lacking clarity” can be refined to mean “not closed under geodesics”—it is possible for the shortest path between two points separated by an axis-parallel split to cross this decision boundary, which can limit the ability of learned splits to generalize.
>
> __Additionally, does the geodesic boundary offer advantages beyond classification scores and interpretation? One potential advantage I am considering is that geodesic trees may require fewer nodes compared to axis-aligned trees to achieve similar classification scores. Could the authors include metrics, such as the average number of nodes used in the experiments presented in Table 1, to support this claim?__
>
> We have added an experiment, detailed in Section A.6.1, on the effect of increasing the maximum depth of the tree on the accuracy of HyperRF, HoroRF, and Euclidean random forests. We found that, controlling for maximum depth, HyperRF was consistently at least as good as the other classifiers, but also that its advantage over other predictors was strongest at lower depths. This supports the theory that, as you suggest, geodesic splits are more parsimonious than Euclidean splits and, interestingly, horospherical splits as well.

---

> > ### Author Response · Authors · 2023-11-22
> > **Thank you for your feedback! (cont)**
> >
> > __For a more comprehensive evaluation, could the authors include additional comparison results? As mentioned in the weaknesses section, there are now several global optimization methods, such as [1], that have demonstrated superior performance compared to CART. It would be valuable to include a comparison with at least one of these methods to illustrate the superior performance of geodesic decision boundaries over axis-aligned boundaries.__
> >
> > We have added a number of additional comparisons to the Appendix. In Section A.6.2, we compare HyperDT to Euclidean and hyperbolic SVMs and logistic regression classifiers; in Section A.6.3, we compare HyperDT and HyperRF to random forests in other models of hyperbolic space; and in Section A.6.5, we test all tree- and forest-based classifiers on a new WordNet classification benchmark. All of these experiments remain favorable to our methods, and hopefully paint a more complete picture of their versatility and effectiveness. Although we did not have time to do a complete comparison to Quant-BnB, we wholeheartedly agree that combining more sophisticated optimization techniques with geodesic decision boundaries is a promising area for future research.
> >
> > __Ablation experiments: The paper lacks ablation studies or experiments to justify the design choices made in the proposed methods, including the selection of candidate hyperplanes. Since the equidistance mid-angles in the proposed method do not have closed-form solutions and rely on numerical solvers, there may be concerns about the algorithm's efficiency. It would be informative to compare the performance and execution times of the equidistance mid-angles with those of naive mid-angles to assess the impact of these choices on the methods' performance and behavior.__
> >
> > We have replaced the numerical approximation of the angular midpoints with a closed-form solution and rewritten Sections 3.3 and A.3 of the paper to reflect this change. The code is updated to use the analytical solution as well. However, we decided an ablation would nevertheless be valuable, as the candidate angle selection is substantially more elaborate than the rest of our algorithm. The procedure and results for this ablation are in Section A.6.6 of the Appendix, and we find that there is a slight reduction in both accuracy and runtime when computing angular midpoints naively using arithmetic means. Because our algorithm is already fairly efficient, we believe it is worthwhile to trade a small increase in runtime for an increase in accuracy.

---

### Official Review · Reviewer_nNVW · 2023-10-30

**Soundness:** 4 excellent
**Presentation:** 3 good
**Contribution:** 3 good
**Rating:** 6
**Confidence:** 3

**Summary:**

This paper presents a extension of traditional Euclidean decision tree algorithms to the hyperbolic space. The proposed method HyperDT is of constant-time decision complexity while mitigating the scalability issues of Euclidean decision tree. The paper further extends HyperDT to random forests tailored for hyperbolic space. The proposed methods HYPERDT and HYPERRF show state-of-the-art accuracy and speed on classification problems compared to existing counterparts on various datasets.

**Strengths:**

1) Extending decision tree to hyperbolic space is of great importance as both hyperbolic space and decision tree algorithm have their own advantages.

2) The proposed method, compared with previous counterparts, maintains constant time decision complexity.

3) A implementation adhering to standard SCIKIT-LEARN API conventions.

4) The presentation and organization of the paper is overall good.

**Weaknesses:**

1) It is unclear why using geodesics as decision boundary is better than using horospheres as used in [1].

2) Although the main goal is to generalzie decision tree to hyperbolic space, the paper lacks some comparisions agaist previous classification methods likes hyperbolic logistic regression and hyperbolic SVM. Without sush comparision, it is not clear whether the proposed hyperbolic decision tree has enough advantages compared with other classifiers.


[1] Lars Doorenbos, et al. Hyperbolic random forests. Arxiv 2023.

**Questions:**

Ref. Weakness

---

> ### Author Response · Authors · 2023-11-22
> **Thank you for your feedback!**
>
> We thank you for your support of our paper and your suggestions for further experiments.
>
> We have added a more theoretical discussion of geodesic boundaries to the introduction, arguing that this is the first random forest method to maintain topological continuity and convexity of all decision areas after an arbitrary number of iterations. We have also performed a number of new experiments, including a comparison to hyperbolic and Euclidean versions of support vector machines and logistic regression classifiers, which is quite favorable to our method as well. We reply to your comments inline:
>
> __It is unclear why using geodesics as decision boundary is better than using horospheres as used in [1].__
>
> We note distinct characteristics of decision boundaries in different classifiers. Euclidean decision trees employ splits that can be extended indefinitely while preserving topological continuity and convexity. In contrast, HoroRF decision regions lack continuity and convexity, especially outside horospheres, where non-convex areas emerge, and the combination of two horospherical splits may result in separate exteriors. Axis-parallel hyperplanes can be extended continuously but lack convexity, allowing geodesics to cross decision boundaries. Geodesic partitions exhibit both convexity under geodesic combinations and topological continuity, aligning more closely with the properties of axis-parallel boundaries in Euclidean classifiers. The superior accuracy of HyperDT and HyperRF, compared to other decision tree-based classifiers, supports the effectiveness of geodesic boundaries in decision trees.
>
> __Although the main goal is to generalzie decision tree to hyperbolic space, the paper lacks some comparisions agaist previous classification methods likes hyperbolic logistic regression and hyperbolic SVM. Without sush comparision, it is not clear whether the proposed hyperbolic decision tree has enough advantages compared with other classifiers.__
>
> We have performed this comparison and wrote it up in Section A.6.2 of the Appendix. These results show HyperDT consistently outperformed both Euclidean and Hyperbolic versions of SVMs and logistic regression classifiers. Since HyperDT is a much simpler classifier than HyperRF, this is strong evidence that our methods outperform other existing hyperbolic classifiers.

---

### Official Review · Reviewer_71Gr · 2023-10-31

**Soundness:** 4 excellent
**Presentation:** 3 good
**Contribution:** 3 good
**Rating:** 8
**Confidence:** 4

**Summary:**

The paper considers the task of creating decision tree models for hyperbolic data. To do so, the splitting rule for axis-aligned (parallel) splits is considered as the function of an inner product and thus a hyperplane. To translate decision trees to the hyperbolic geometry, the equivalent inner product and hyperplanes are used to define a hyperbolic decision boundary. This approach is utilized to create a random forest-style algorithm and is tested on various hyperbolic datasets.

**Strengths:**

- The visual presentation is very good. Figures 1 & 2 provide a nice intuition on how axis-align splits are generalized to the hyperbolic regime.
- The simplicity of the approaches mechanism for lifting decision trees to hyperbolic geometry allows for an efficient alternative than prior baselines (further explored in the experimental setting).
- Competitive empirical results showing its practicality.

**Weaknesses:**

- Some sections on parameterizing decision boundaries are unclear. (See questions below)
- It maybe worth adding explicit clarification in the contributions that the data input for HyperDT are hyperbolic data / representations, ie, a separate embedding process is needed to apply such a model to Euclidean data.

**Questions:**

"Parameterizing Decision Boundaries"

I find that the stated motivation and reasoning for Eq. (11) & (12) unclear. This further extends to the corresponding appendix sections:

- Language in the description of Eq. (11) seems imprecise. "... the hyperplane parameterized by $d$ and $\theta$ will intersect $\mathbb{H}^D$ at: [ Eq.(11)]". This seems to suggest there is only one point of intersection. I assume that the point characterized by Eq (11) is the closest point to the origin in said intersection?
- As a general suggestion, it may be useful to the reader to have both Eq. (7) and Eq. (11) in Figure 1.
- In the set up of Appendix A.2, a vector $\mathbf{u}$ is defined. It is unclear exactly what this is. From the combined definition in A.2.2 and Eq. (24), I believe that it is a unit vector pointing in a spacelike direction? What exact is its role?
- I am unsure how you went from Eq. (24) to Eqs. (25-27). From what I am guessing, $\mathbf{u}$ defines the direction of the intersecting geodesic, but I am unsure about the "\sinh" parameterization given.
- What is the connection between dimension $d'$ in Eq. (12) and the normal vector characeterization of Eq. (8/9)?

Other questions:

- Although the construction of the decision boundaries follows from their Euclidean axis-aligned counterparts, can the construction be generalized for any hyperplane (intersecting with the $\mathbb{H}^D$)? How would this change generating the decision boundary? Especially due to fact that the assumption of letting dimension except for 0 and d being zero is used to generate the hyperplane characterization.
- Following from the above, why is Eqs. (25-27) restricted to having the geodesic follow an axis $d'$?

Minor:
- Eq. (10), "cos" missing "\cos"

---

> ### Author Response · Authors · 2023-11-22
> **Thank you for your feedback!**
>
> We thank you for your support of our method and your close engagement with the mathematical details of our model.
>
> We have taken your review into consideration and rewritten the sections of the paper and appendix dealing with parameterizing the decision boundaries. We hope that the new version is clearer, more detailed, and more general. In particular, we explicitly show how to parameterize decision boundaries to arbitrary dimensions. Please see sections 3.4 and A.2 for the fully revised description. We have also replaced the numerical approximation for midpoint angles with an analytic closed form, which is described in sections 3.3 and A.3. We will now respond to your comments inline:
>
> __Language in the description of Eq. (11) seems imprecise. "... the hyperplane parameterized by and will intersect at: [ Eq.(11)]". This seems to suggest there is only one point of intersection. I assume that the point characterized by Eq (11) is the closest point to the origin in said intersection?__
>
> (Please note that this is now equation 12 in the paper.)
>
> You are correct to point out that we are choosing the closest point for this analysis, which can be done by setting all other dimensions to 0. This additionally reduces the computation to a 1-dimensional hyperbola, as shown in Figure 4 in the Appendix. We have changed the language in the paper and appendix to be clearer about this by adding the phrases “when all other dimensions are 0…” and “note that this is the point on the intersection of $\mathbf{P}(\theta)$ and $\mathbb{H}^{D,K}$ closest to the origin.”
>
> We further hope that, by making the role of other dimensions (and specifically the u vectors) clearer, it will be clearer why it is natural to start parameterizing the geodesic submanifold by choosing the closest point to the origin.
>
> __As a general suggestion, it may be useful to the reader to have both Eq. (7) and Eq. (11) in Figure 1.__
>
> Because Equations 7 and 11 describe the specific axis-inclined hyperplanes used in HoroDT, whereas we wish for Figure 1 to deal with geodesic separators more generally, we have elected to keep Figure 1 as is. While it is true that the particular hyperplane drawn in Figure 1 is axis-inclined (hence the theta), we wish to keep the figure general to provide context and intuition for geodesics in general.
>
> __In the set up of Appendix A.2, a vector is defined. It is unclear exactly what this is. From the combined definition in A.2.2 and Eq. (24), I believe that it is a unit vector pointing in a spacelike direction? What exact is its role?__
>
> (Equation 24 has been removed; it is now broken into Equations 30-32.)
>
> We hope that our rewritten Section A.2 is much clearer with respect to the basis vectors of the decision hyperplane. Indeed, we choose the $\mathbf{u}$ vectors to be unit vectors in each of the non-$d$ spacelike dimensions (see the new Eq. 21). Section A.2.1 describes this basis.
>
> Spacelike unit vectors are used in the construction of geodesic arcs according to the method described in section A.2.3, and can be further composed into geodesic submanifolds according to the method in A.2.4.
>
> __I am unsure how you went from Eq. (24) to Eqs. (25-27). From what I am guessing, defines the direction of the intersecting geodesic, but I am unsure about the "\sinh" parameterization given.__
>
> (Equation 24 is now broken into Equations 30-32; Equations 25-27 are now 33-35)
>
> Here are two links that lay out a simple version of this formulation of the geodesic:
>
> https://math.stackexchange.com/questions/4663901/geodesics-in-hyperboloid-model-of-hyperbolic-space#:~:text=being%20the%20formula%20for%20a,the%20definition%20of%20the%20hyperboloid
>
> https://en.wikipedia.org/wiki/Hyperboloid_model#Straight_lines
>
> We have added a citation to Chami et al (2019): Hyperbolic Graph Convolutional Networks before Equations 33-35 as well: Proposition 3.1 in this paper contains a version of this formulation.
>
> __What is the connection between dimension in Eq. (12) and the normal vector characeterization of Eq. (8/9)?__
>
> (Equation 12 has been removed. It is now Equations 13-16).
>
> The dimension $d$ used in the normal vector characterization and the dimension $d$ used in the characterization of the geodesic are the same: the basis vector $\mathbf{v}$ is nonzero in precisely dimensions 0 and $d$, which corresponds to the decision hyperplane $P(\theta)$ being inclined along axis $d$.

---

### Official Review · Reviewer_RLmq · 2023-11-01

**Soundness:** 3 good
**Presentation:** 3 good
**Contribution:** 3 good
**Rating:** 6
**Confidence:** 2

**Summary:**

This paper proposes a decision tree algorithm: HYPERDT. It leverages inner products to adapt Euclidean decision tree algorithms to hyperbolic space by characterizing Euclidean decision tree algorithms in terms of inner products, offering a natural modification of these algorithms to hyperbolic space. This approach works for all negative curvatures and eliminates the necessity for pairwise comparisons between data points. HYPERDT eliminates the need for computationally intensive Riemannian optimization, numerically unstable exponential and logarithmic maps, or pairwise comparisons between points by leveraging inner products to adapt Euclidean decision tree algorithms to hyperbolic space.

**Strengths:**

1. Novelty: The paper introduces a novel approach to decision tree algorithms by extending them into hyperbolic space. This is a new and innovative idea that has not been explored extensively in the machine learning community.

2. Performance: The authors demonstrate that hyperbolic decision trees and random forests can outperform their Euclidean counterparts in certain scenarios. This suggests that hyperbolic geometry has the potential to improve the performance of machine learning algorithms.

3. Clarity: The paper is well-written and easy to understand, even for readers who are not familiar with hyperbolic geometry. The authors provide clear explanations of the concepts and algorithms presented in the paper.

**Weaknesses:**

However, I have the following concerns:
1. All datasets seem to be better suited for the algorithm proposed by the authors. The authors do not provide the performance of the algorithm for other different types of datasets.
2. How about the deeper tree (T>3 or even 7)?
3. What is the scalability of the algorithm? The author only tested the datasets of samples within 1000 and dimensions within 16.
4. There are many other new decision tree training algorithms proposed these years. What is the performance of HYPERDT when compared with other advanced decision tree algorithms? (e.g., [1], [2])

[1] McTavish, H., Zhong, C., Achermann, R., Karimalis, I., Chen, J., Rudin, C., & Seltzer, M. (2022, June). Fast sparse decision tree optimization via reference ensembles. In Proceedings of the AAAI Conference on Artificial Intelligence (Vol. 36, No. 9, pp. 9604-9613).

[2] Lin, J., Zhong, C., Hu, D., Rudin, C., & Seltzer, M. (2020, November). Generalized and scalable optimal sparse decision trees. In International Conference on Machine Learning (pp. 6150-6160). PMLR.

**Questions:**

Please refer to Strengths and Weaknesses.

---

> ### Author Response · Authors · 2023-11-22
> **Thank you for your feedback!**
>
> We thank you for your endorsement of our method’s novelty and performance, your appreciation of our paper’s presentation, and your thoughtful suggestions for further experiments.
>
> We have carried out all experiments you have proposed and added them to Section A.6, “Additional Experiments,” in the appendix. We now address your comments inline:
>
> __All datasets seem to be better suited for the algorithm proposed by the authors. The authors do not provide the performance of the algorithm for other different types of datasets.__
>
> In Section A.6.5, we have added a new comparison where we classify hyperbolic embeddings of WordNet data, following the example in the HoroRF paper. The data for this was not public at submission time, so we are happy to have the opportunity to extend our analysis to a problem from natural language processing. The results of this experiment once again favor HyperDT/RF over HoroDT/RF and Scikit-Learn’s Euclidean classifiers.
>
> Additionally, we would look to draw your attention to Section A.6.4 on classifying hyperbolic image embeddings, which was included in the Appendix to the original submission as its own section. This experiment shows mixed results: although HyperRF with hyperbolic embeddings outperforms all other combinations of random forest algorithm and geometry, we fail to meet the logistic regression baseline reported in the original paper. This suggests that random forests in general are not well-suited to the image classification task, but HoroRF is nevertheless more effective than its Euclidean counterparts.
>
> __How about the deeper tree (T>3 or even 7)?__
>
> We have performed a number of new scaling benchmarks, which we report in Section A.6.1. Figure 6d shows the effect of tree depth (up to 20) on the runtime and accuracy of HyperRF; Figure 7 compares the accuracy scaling with HoroRF and scikit-learn random forests. In particular, we are able to demonstrate a consistent advantage over the other methods at small tree depths, with all classifiers converging to roughly the same accuracy at high depths. We would like to emphasize that these results are on a toy dataset (mixture of Gaussians), and overfitting may become a bigger problem for all classifiers on other datasets.
>
> __What is the scalability of the algorithm? The author only tested the datasets of samples within 1000 and dimensions within 16.__
>
> Section A.6.1 now contains runtime and accuracy benchmarks for increasing the number of samples up to 3,000, the dimensionality up to 64, the number of trees in an ensemble up to 30, and the maximum depth up to 20. With the exception of tree depth, each experiment is consistent with linear scaling of runtime; interestingly, runtime seems to scale sublinearly with tree depth. This is likely due to two factors: first, the maximum depth is much higher than the actually achieved depth due to early stopping conditions of the algorithm; second, caching optimizations that allow us to reuse computations during the candidate selection step of the CART algorithm speed up late splits.
>
> __There are many other new decision tree training algorithms proposed these years. What is the performance of HYPERDT when compared with other advanced decision tree algorithms? (e.g., [1], [2])__
>
> While the duration of the reply period limited us to adding comparisons against hyperbolic logistic regression and hyperbolic SVMs, HyperDT can be adapted to other decision tree algorithms besides CART. The decision to adapt CART was motivated by its widespread popularity, effectiveness in machine learning, and the desire to avoid confounding the comparison of predictors’ geometries by differences in optimization procedures. We have updated the related work section and conclusion to include a discussion of other decision tree frameworks and leave additional extensions for future work.

---

### Official Review · Reviewer_ZUYY · 2023-11-01

**Soundness:** 3 good
**Presentation:** 4 excellent
**Contribution:** 4 excellent
**Rating:** 8
**Confidence:** 3

**Summary:**

The paper presents an extension of standard decision trees to the data situated in hyperbolic spaces. Certain types of data are better modeled by sets of recursive splits (decision trees), and also currently there is a growing attention to data analysis in hyperbolic spaces, due to development of hyperbolic embedding models. While there was HoroRF, a way to recursively split data in hyperbolic space, the proposed approach uses a different spliting model that is much simpler to operate and train, and this finding contains a valuable theoretical insight. Practically it also outperforms previously available methods.

**Strengths:**

The paper significantly advances the sub-field of decision tree models for hyperbolic spaces by presenting a novel decision tree model, that produces the splits in a different way, compared to previous state of the art, and leads to asymptotically faster computation times and better practical evaluation.

**Weaknesses:**

What is lacking in my opinion is a theoretical analysis of different decision boundaries and their performance for certain kinds of data (certain kinds of distributions of the data, that we may consider natural in certain classes of problems). For example, in case of regular decision trees, it may be observed that tabular data (where regular decision trees are best performers) is usually situated on axis-parallel hyperplanes and is easily separated by similar axis-parallel splits (due to naturally repeating nature / distribution of the data in tables along some dimensions)

The paper contains an interesting (and possibly simple to further analyze) case of synthetic data in hyperbolic space consisting of a sample from a mixture of Gaussians. It was observed empirically, that proposed method consistently outperforms the previous HoroRF/HoroDT. Providing a theoretical analysis of why it is the case would fill in the gaps and justify a choice of the proposed subset among all possible decision boundaries in hyperbolic space.

**Questions:**

- Considering the proposed method and HoroRF, we know two ways of splitting the hyperbolic space, which correspond to an inductive bias that we impose on the data. Are there any other inductive biases / splits that can be considered, but perhaps are currently infeasible or otherwise unsuitable?
- Particularly, there is an inherent asymmetry of the hyperbolic space towards the center of the Poincare ball - and embeddings usually automatically (without being instructed to do so, during regular optimization) utilize this asymmetry to represent well the hierarchical data. Can we use this asymmetry in designing the splitting method?
- Following from the previous question, consider the following baseline (should be worse than directly working in hyperbolic space, but better than the straightforward use of regular decision trees):
-- Transform the data into the Poincare ball. After that, ignore the hyperbolic nature of the data (as in regular decision trees) and transform the data into spherical coordinate system. Now run the regular decision tree on such data.
-- In this baseline the splits will be concentric spheres which are further split into spherical sectors and sub-sectors.
-- Would be nice to see a comparison between proposed approach and some approach (such as suggested above) that lies in between the Hyperbolic Trees and Regular Decision Trees.

---

> ### Author Response · Authors · 2023-11-22
> **Thank you for your feedback!**
>
> Thank you very much for your feedback and your endorsement of our paper’s contribution to the field.
>
>
> We have revised the paper in accordance with your comments. Specifically, we have added a discussion of the advantages of geodesic decision boundaries to the Introduction, Methods, and Conclusions, and added experiments involving random forests in other models of hyperbolic geometry to the appendix. Our point-by-point response to your comments follows:
>
>
> __What is lacking in my opinion is a theoretical analysis of different decision boundaries and their performance for certain kinds of data (certain kinds of distributions of the data, that we may consider natural in certain classes of problems). For example, in case of regular decision trees, it may be observed that tabular data (where regular decision trees are best performers) is usually situated on axis-parallel hyperplanes and is easily separated by similar axis-parallel splits (due to naturally repeating nature / distribution of the data in tables along some dimensions)__
>
> The paper contains an interesting (and possibly simple to further analyze) case of synthetic data in hyperbolic space consisting of a sample from a mixture of Gaussians. It was observed empirically, that proposed method consistently outperforms the previous HoroRF/HoroDT. Providing a theoretical analysis of why it is the case would fill in the gaps and justify a choice of the proposed subset among all possible decision boundaries in hyperbolic space.
>
>
> We have added a brief discussion of the advantages of geodesic decision boundaries to the Introduction of the paper. Our position is that the decision boundaries employed by Euclidean CART algorithms partition the space into continuous and convex subregions (“decision areas”), but these properties do not carry over to the hyperbolic case when applying existing tools. Specifically, HoroRF splits the space into one convex and one nonconvex region, and further splitting the nonconvex regions can create non-continuous decision areas. Euclidean random forests, on the other hand, preserve continuity but are not closed under geodesic combinations of their elements. Ours is the first method to guarantee both continuity and convexity of the resulting decision areas in hyperbolic space, which we believe helps the decision areas more naturally follow the data distribution and to generalize better with fewer splits. The empirical performance of our methods on benchmarks, as well as the advantages of Klein model embeddings over other models of hyperbolic space with Euclidean classifiers (see Section A.6.3 in the Appendix), corroborate the advantages of classification by geodesic splits.
>
>
> __Considering the proposed method and HoroRF, we know two ways of splitting the hyperbolic space, which correspond to an inductive bias that we impose on the data. Are there any other inductive biases / splits that can be considered, but perhaps are currently infeasible or otherwise unsuitable?__
>
>
> Another simple split to consider, an unrestricted linear split, is used by support vector machines and logistic regression classifiers. This type of split also has no geometric interpretation in the hyperboloid model, but it has the advantage of being able to search over all possible hyperplanes. The experiment carried out in Section A.6.2 suggests that our classifier works better, although it is not a 1:1 comparison because we permit a maximum depth of 3. The comparisons to other hyperbolic models in Section A.6.3 are also new classes of splits, although (with the exception of the Klein model) these hyperplane splits of the coordinate system are not immediately meaningful. Further comparisons to oblique decision trees and kernel support vector machines may also be informative.
>
>
> __Particularly, there is an inherent asymmetry of the hyperbolic space towards the center of the Poincare ball - and embeddings usually automatically (without being instructed to do so, during regular optimization) utilize this asymmetry to represent well the hierarchical data. Can we use this asymmetry in designing the splitting method?__
>
>
> There is a similar asymmetry to the hyperboloid model, where deeper levels of the hierarchy are represented as having a higher value in the timelike dimension; this is equivalent to having a greater radius in the Poincare disk model. In general, HyperDT/RF splits utilize this by having angles closer to $\pi/4$ or $3\pi/4$, corresponding to planes that intersect the hyperboloid farther up the timelike axis. Evidence from the new classification benchmark on nested WordNet embeddings in Section A.6.5 in the Appendix suggests that the splits used in HyperDT/RF indeed capture multiple levels of the hierarchy successfully compared to both HoroDT/RF and Euclidean decision trees/random forests, even in nested multiclass classification settings.

---

> > ### Author Response · Authors · 2023-11-22
> > **Thank you for your feedback! (cont)**
> >
> > __Following from the previous question, consider the following baseline (should be worse than directly working in hyperbolic space, but better than the straightforward use of regular decision trees): -- Transform the data into the Poincare ball. After that, ignore the hyperbolic nature of the data (as in regular decision trees) and transform the data into spherical coordinate system. Now run the regular decision tree on such data. -- In this baseline the splits will be concentric spheres which are further split into spherical sectors and sub-sectors. -- Would be nice to see a comparison between proposed approach and some approach (such as suggested above) that lies in between the Hyperbolic Trees and Regular Decision Trees.__
> >
> >
> > We have added an experiment to Section A.6.3 of the Appendix extending our benchmarks to Euclidean decision tree and random forest classifiers to the hyperboloid, Klein, Poincare, and Euclidean (by application of the logarithmic map at the origin) coordinate systems. These experiments paint a more complete picture of the (in)compatibility between Euclidean decision boundaries and hyperbolic embeddings of various sorts. It is particularly interesting to note that Klein model embeddings generally outperformed other models of hyperbolic space, likely because geodesics are represented by straight lines in the Klein model. However, the only statistically significant advantages we observed consistently were between HyperDT/RF and Euclidean classifiers in the hyperboloid model.

---

> > > ### Comment · Reviewer_ZUYY · 2023-11-30
> > >
> > > Thank you for a detailed answer to all my questions and a quick confirmation of the claims by additional experiments.
> > > Your note and the discussion in the updated version of the paper about a property of good decision regions being convex and continuous well justifies the proposed choice of the considered set of decision boundaries.
> > >
> > > I only want to note that the added experiments in Appendix A6.2, A6.3 that explore other decision boundaries - are based on an "easy" dataset, where even a regular Decision Tree and also the Logistic Regression achieve > 99% accuracy in dimensions higher than 2. Therefore the  particular conclusions from A6.2 and A6.3 may not generalize well to other conditions.  I think this could be resolved by reproducing in A6.2 and A6.3 the full Table 1 (with new models considered) from Section 4.4 that contains harder datasets.
> > >
> > > At the same time, other parts of the paper provide enough evidience that the proposed approach is sound.
> > >
> > > Overall, the updated version agrees with my initial positive evaluation of the paper.

---

### Author Response · Authors · 2023-11-22
**General response**

We thank the reviewers for their helpful feedback and thoughtful commentary. In particular, there was consensus that our paper was novel, technically sound, clearly written, and has potential for significant impact.


Per reviewers’ comments, we have also made the following improvements to the paper:
* We have added some discussion of why geodesic decision boundaries are theoretically preferable to other decision boundaries, e.g. horospheres and axis-aligned hyperplanes. Specifically, we stipulate that topological continuity and convexity of decision areas—properties we tend to take for granted in Euclidean classifiers—are important but not guaranteed by other methods. This discussion appears in the Introduction, Section 3.2 “Extension to the hyperboloid model,” and the Conclusions section.
* We have added a number of new experiments to the appendix, and an “additional experiments” section to the Results describing them. These experiments present further evidence that HyperDT and HyperRF are accurate and scalable:
  * In Section A.6.1, we show that HyperRF runtime scales linearly with number of data points, dimensions, and number of predictors, and even sublinearly with maximum tree depth. We also show that HyperRF’s accuracy advantage over HoroRF and Euclidean random forests persists across maximum depth values, but is strongest for a low number of splits.
  * In Section A.6.2, we show that HyperDT consistently outperforms Euclidean and hyperbolic versions of support vector and logistic regression classifiers.
  * In Section A.6.3, we show that HyperRF is generally better than Euclidean random forests run on other models of hyperbolic geometry, but Poincare and Klein models are better suited to Euclidean random forests than the hyperboloid representation.
  * We have moved the MERU experiment from the previous submission to Section A.6.4 of the appendix.
  * In Section A.6.5, we extend our analysis to classifying hyperbolic embeddings of WordNet, again beating the other methods.
  * In Section A.6.6, we show the effects of replacing the complicated angular midpoint computation we describe in Section 3.3 of the paper with a naive angular bisection and find that this ablation hurts accuracy but slightly improves runtime.
* We rewrote mathematical sections of the paper and appendix. We believe these rewrites are substantially clearer, more rigorous, and more general than the previous draft:
  * _Parameterizing geodesic submanifolds_: sections 3.4 and A.2 build up the mathematical framework for describing geodesic manifolds in more detail. Our description now extends to geodesic submanifolds of arbitrary dimension, rather than simply geodesics in 2-dimensional hyperbolic space.
  * _Angular midpoint formulas_: we rewrote sections 3.3 and A.3, replacing a numerical approximation for the angular midpoint between two decision boundaries with a closed-form analytic solution. We implemented this change in code, which resulted in an appreciable speed-up in our runtime. As such, we have also updated Section 4.4 “Runtime analysis” and Figure 3 to reflect the new runtimes.
* We have added a discussion of non-CART based, globally optimal algorithms to the Related Work and Conclusion sections of the paper, emphasizing the combination of hyperbolic decision boundaries with better optimization techniques as a promising area for future research. We believe that more advanced decision tree algorithms, being based on Euclidean splits, can be modified using the same method we propose in this paper for CART.


We have additionally made many small fixes to the phrasing of various concepts in the manuscript. In particular, we have shortened much of the text to keep the paper within the page limit after adding the aforementioned changes.

---

### Meta-Review · Area_Chair_qBta · 2023-12-15

**Metareview:**

An interesting paper on learning and embedding the boundaries of decision tree classification in (after rebuttal) several models of hyperbolic geometry, as noted in particular by two reviewers, using CART as "master" training routine. It is an interesting way to represent the domain, the recursive splitting of the tree and see how the data is positioned with respect to those splits.

Some reviewer clearly mention that a more formal approach is needed (ZUYY, uhLf), but at the same time reviewers praise the applicability of the paper via its alignment w/ scikit-learn conventions. It is important to add the new experiments to the paper. Also, the authors have made a substantial effort to align with formatting comments of reviewers (e.g. 71Gr) and have substantially ramped up experimental results at rebuttal time.

**Justification For Why Not Higher Score:**

Reviews of nNVW, uhLf, in particular the mention of suboptimality of CART, to which the authors somehow lazily replied that "other algorithms" (no justification) can be used and CART is popular. The former answer is borderline acceptable, the latter one disputable.

**Justification For Why Not Lower Score:**

Reviewer ZUYY, 71Gr gave the paper a 8 rating with good arguments in their review.

---

### Decision · Program_Chairs · 2024-01-16

Accept (poster)